# *Staphylococcus aureus* induces an itaconate-dominated immunometabolic response that drives biofilm formation

Kira L. Tomlinson [1,7], Tania Wong Fok Lung [1,7], Felix Dach [1,2,7], Medini K. Annavajhala [3], Stanislaw J. Gabryszewski[1], Ryan A. Groves[4], Marija Drikic[4], Nancy J. Francoeur[5], Shwetha H. Sridhar[5], Melissa L. Smith[5], Sara Khanal[6], Clemente J. Britto[6], Robert Sebra[5], Ian Lewis [4], Anne-Catrin Uhlemann[3], Barbara C. Kahl [2], Alice S. Prince [1] & Sebastián A. Riquelme [1✉]

*Staphylococcus aureus* is a prominent human pathogen that readily adapts to host immune defenses. Here, we show that, in contrast to Gram-negative pathogens, *S. aureus* induces a distinct airway immunometabolic response dominated by the release of the electrophilic metabolite, itaconate. The itaconate synthetic enzyme, IRG1, is activated by host mitochondrial stress, which is induced by staphylococcal glycolysis. Itaconate inhibits *S. aureus* glycolysis and selects for strains that re-direct carbon flux to fuel extracellular polysaccharide (EPS) synthesis and biofilm formation. Itaconate-adapted strains, as illustrated by *S. aureus* isolates from chronic airway infection, exhibit decreased glycolytic activity, high EPS production, and proficient biofilm formation even before itaconate stimulation. *S. aureus* thus adapts to the itaconate-dominated immunometabolic response by producing biofilms, which are associated with chronic infection of the human airway.

---

[1] Department of Pediatrics, Columbia University, New York, NY 10032, USA. [2] Institute of Medical Microbiology Münster, University Hospital, Münster 48149, Germany. [3] Department of Medicine, Columbia University, New York, NY 10032, USA. [4] Department of Biological Sciences, University of Calgary, Calgary T2N 1N4, Canada. [5] Department of Genetics and Genomic Sciences, Mt. Sinai Icahn School of Medicine, New York, NY 10029, USA. [6] Section of Pulmonary, Critical Care, and Sleep Medicine, Yale University School of Medicine, New Haven, CT 06520, USA. [7] These authors contributed equally: Kira L. Tomlinson, Tania Wong Fok Lung, Felix Dach. ✉email: sr3302@cumc.columbia.edu

**S**taphylococcus aureus is a major Gram-positive pathogen associated with a diverse array of acute and chronic infections[1]. The lung is a common site of S. aureus infection, as evidenced a decade ago by the epidemic of acute pneumonia caused by a toxin-producing methicillin-resistant S. aureus (MRSA) strain[2,3]. More commonly, however, S. aureus causes chronic lung infections in the setting of damaged airways, including ventilator associated pneumonia and pneumonia in chronic obstructive pulmonary disease (COPD) and cystic fibrosis (CF)[1]. To persist in the airway, these organisms must rapidly adapt to the brisk immune response in the local environment, which is dominated by resident and recruited immune cells and their bactericidal products[4,5]. In this era of multidrug resistance and intractable S. aureus infections, it is critical to understand how these organisms adapt to this inflammatory pressure and persist.

During the inflammatory response to lipopolysaccharide (LPS), a major virulence factor of Gram-negative bacteria, there is substantial release of succinate, reactive oxygen species (ROS), and itaconate into the airway[6,7]. Succinate and ROS activate inflammation via stabilization of hypoxia-induced factor 1α (HIF-1α) and induction of the potent anti-bacterial cytokine IL-1β[8]. This inflammatory environment provides selective pressure on bacteria[9], inducing genotypic and phenotypic changes that can be complemented by itaconate[10]. Itaconate is produced by Immune-Responsive Gene 1 (IRG1), and suppresses inflammation by inhibiting succinate oxidation by succinate dehydrogenase[11]. Itaconate also limits inflammation by blocking macrophage metabolic activity through inhibition of the glycolytic enzymes aldolase and glyceraldehyde-3-phosphate dehydrogenase (G3PDH)[12,13]. Due to its electrophilicity, itaconate has several other downstream effects, including activation of Nrf2-dependent and independent anti-oxidant programs that prevent excessive tissue damage[14]. In addition, itaconate is bactericidal against intracellular pathogens like Legionella pneumophila and S. aureus[15,16]. Thus, the relative abundance of itaconate, ROS, and succinate, and the ability of the aspirated pathogens to respond to these metabolites are important factors in the pathogenesis of pneumonia.

Metabolic flexibility is a major feature of S. aureus pathogenesis, enabling proliferation or at least persistence in seemingly adverse environmental niches[17]. While Gram-negative bacteria such as Pseudomonas aeruginosa and Salmonella enterica metabolize succinate and itaconate[9,10,18], S. aureus does not[9,16,19], and instead prefers glucose as its main carbon source[20–22]. S. aureus catabolizes glucose to rapidly generate ATP for planktonic proliferation, and this process helps to establish acute infection in tissues like the skin[20]. However, glucose can also be used for the generation of biofilm[23,24], which is an especially important growth modality that supports a longstanding bacterial community that is protected from phagocytosis[25] and antibiotics[19,26,27]. A major component of staphylococcal biofilms are extracellular polysaccharides (EPS), which are synthesized from glucose units that can be imported from the environment or produced through gluconeogenesis[28]. Therefore, while biofilm acts as an oxidant sink, it also induces a metabolic program that itself generates less endogenous oxidant stress, at the expense of glycolytic metabolism and planktonic proliferation. Having observed substantial accumulation of itaconate in the airways of subjects that develop recurrent and chronic S. aureus infection[10], we postulated that this electrophile might be a major stimulus of the biofilm lifestyle that protects S. aureus against oxidant stress and enables persistent infection[29–31].

In the studies presented in this report, we demonstrate that S. aureus infection does not induce succinate release but robustly stimulates IRG1 and itaconate production in the airway through induction of host mitochondrial ROS. Acute exposure to itaconate leads to transcriptomic and metabolic changes in S. aureus, characterized by decreased glycolysis coupled with the selection of strains that exhibit increased carbon flux through gluconeogenesis to produce biofilm. These changes were recapitulated in a longitudinal collection of S. aureus clinical isolates that were chronically exposed to airway immunometabolites. Itaconate thus imposes a metabolic pressure on S. aureus, which selects strains that produce EPS at the expense of proliferation, enabling survival of a biofilm-forming bacterial community that can persist for years in the human lung.

## Results

**S. aureus infection specifically induces airway itaconate production.** To evaluate the immunometabolic response to Gram-positive bacteria we infected mice intranasally with S. aureus (USA300-LAC) and performed metabolomics on the bronchoalveolar lavage (BAL) fluid. As a positive control, we used an LPS-expressing Pseudomonas aeruginosa strain (PAO1)[10]. We observed that S. aureus induced a 6-fold increase in the amount of airway itaconate (from 18.6 nM to 119 nM in the diluted BAL fluid), without a concomitant increase in succinate, α-D-glucose or fumarate, which differed from the immunometabolic response to the same dose of P. aeruginosa (Fig. 1a)[10]. This unique airway metabolic profile correlated with the cytokine response, as S. aureus induced less IL-1β and IL-6 than P. aeruginosa, but similar TNFα (Fig. 1b). The itaconate-dominated response induced by LAC also correlated with increased IRG1 expression in alveolar macrophages (Fig. 1c, d), suggesting that resident myeloid cells are a source of itaconate during S. aureus infection. To confirm the clinical relevance of these findings, we infected mice with a previously described[19] set of S. aureus clinical isolates obtained from a chronically infected CF patient (A1 (2013) and A5 (2017) from sputum; A6 (2017) from blood) and found that these isolates also induced itaconate accumulation in BAL fluid (Fig. 1e). Interestingly, the A6 blood isolate induced less itaconate than the sputum-derived strains, suggesting that different biological niches select for bacteria with varying capacity to trigger itaconate production. We also analyzed sputum from 5 healthy subjects and 7 S. aureus chronically infected CF individuals, and observed that the latter contained more itaconate (Fig. 1f). This itaconate accumulation occurred in individuals infected with S. aureus alone, as well as in individuals co-infected with other pathogens, like P. aeruginosa (Fig. 1f).

Given that itaconate is synthesized to limit the generation of oxidative species in LPS-primed macrophages[14], we predicted that itaconate generated during S. aureus infection represents a similar antioxidant response. We confirmed that infection with LAC induced production of mitochondrial anion superoxide $O_2^{*-}$ (hereafter referred to as mitochondrial ROS) in both human THP-1 cells (Fig. 1g, h) and mouse bone-marrow-derived macrophages (BMDMs; Supplementary Fig. 1a, b). We then demonstrated that treatment with a specific mitochondrial ROS scavenger (MitoTEMPO) reduced IRG1 expression in THP-1 cells (Fig. 1i), suggesting that itaconate synthesis is a response to mitochondrial $O_2^{*-}$. In parallel, we confirmed that type I interferons, which are also associated with itaconate synthesis in the presence of LPS[32], were not linked to itaconate production by S. aureus, as these cytokines did not accumulate in the murine airway during infection (Supplementary Fig. 2a). Together, these data suggest that S. aureus infection induces expression of IRG1 and itaconate synthesis as an anti-oxidant response in myeloid cells.

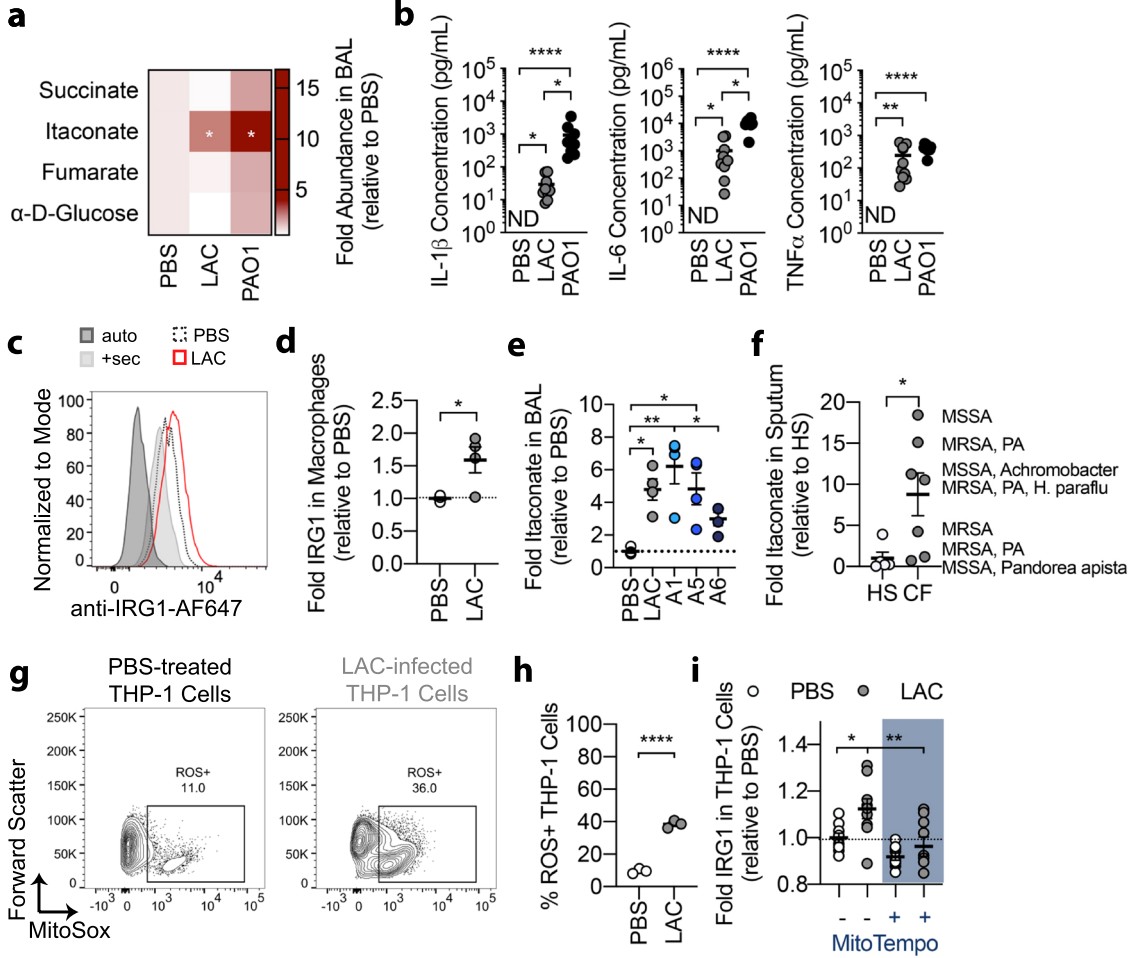

**Fig. 1 S. aureus induces itaconate release in the airway. a** Bronchoalveolar lavage (BAL) fluid metabolites and **b** cytokines from mice treated with PBS or infected with LAC (*S. aureus* USA300) or PAO1 (*P. aeruginosa*). **c**, **d** Irg1 expression in the alveolar macrophages of PBS-treated or LAC-infected mice. **e** BAL itaconate from mice infected with the *S. aureus* clinical isolates A1, A5, A6. **f** Sputum itaconate from healthy subjects (HS) or CF patients (CF). **g**, **h** Mitochondrial ROS generation in PBS-treated and LAC-infected THP-1 cells. **i**. IRG1 expression in PBS-treated and LAC-infected THP-1 cells treated with a mitochondrial ROS scavenger (MitoTempo) or vehicle (PBS). Data are shown as mean ± SEM from $n = 4$ mice (**a**, **e**), 9 mice (**b**), 3 mice (**c**, **d**, **g**, **h**), or 9 biological replicates from 3 independent experiments (**i**). Significance determined by Two-Way ANOVA with Dunnett's Multiple Comparisons (**a**), Kruskal-Wallis (**b**), two-tailed t-Student (**d**, **f**, **h**), or One-Way ANOVA with Tukey's Multiple Comparisons (**e**, **i**); *$P < 0.05$; **$P < 0.01$; ****$P < 0.0001$.

**S. aureus glycolysis induces host mitochondrial ROS and itaconate production**. The metabolic profile preferred by *S. aureus* at specific sites of infection directs the type of immune response deployed by the host[33,34]. We postulated that the ROS-response induced by LAC was associated with bacterial glycolysis, as we previously observed in skin[35,36]. We infected THP-1 cells with WT LAC or a pyruvate kinase (Δ*pyk*) mutant, which had impaired glycolytic activity (Fig. 2a, left panel) and glucose-dependent oxygen consumption (Fig. 2a, right panel). While the WT strain reduced THP-1 mitochondrial membrane potential and induced mitochondrial ROS production, the Δ*pyk* mutant did not (Fig. 2b–d). These findings were recapitulated in vivo, as alveolar macrophages from WT LAC-infected mice exhibited greater ROS production than those from Δ*pyk*-infected mice (Fig. 2e,f). This decreased ROS induction correlated with diminished itaconate levels in the BAL fluid of the Δ*pyk*-infected mice (from 193 nM for LAC to 103 nM for Δ*pyk* in the diluted BAL fluid) (Fig. 2g), supporting the conclusion that *S. aureus* glycolytic activity stimulates itaconate production through myeloid mitochondrial stress. Importantly, this induction of host itaconate production was not dependent on the production of lactate, a fermentative metabolite downstream of glycolysis that

induces immune cell activation[37], as a lactate-deficient mutant (Δ*ddh/ldh1/ldh2*) induced similar levels of itaconate as WT LAC in the murine airway (Supplementary Fig. 2b).

The distinct immunometabolic responses to WT and Δ*pyk* LAC could not be explained by significant differences ($P > 0.05$) in colony counts (Fig. 2h). Even at a later time point (40 hours post infection), the WT strain and Δ*pyk* mutant were present at similar levels in the lung (Supplementary Fig. 2c), suggesting that glycolysis is not essential for *S. aureus* survival during acute lung infection. We did not observe a significant correlation between BAL CFUs and itaconate accumulation (Pearson $r = −0.2075$, $p = 0.4405$), again suggesting that the airway *Irg1* response is triggered by staphylococcal metabolism and not bacterial load (Supplementary Fig. 2d). While *S. aureus* glycolytic activity correlated with differences in airway cytokine production, including less induction of ROS-dependent IL-1β and IL-6 by the Δ*pyk* mutant (Supplementary Fig. 2e), there were no differences in numbers of resident and recruited immune cells (Fig. 2i). These data indicate that staphylococcal glycolytic metabolism influences the airway immune-metabolic response during infection, stimulating host mitochondrial ROS, itaconate, and cytokine production.

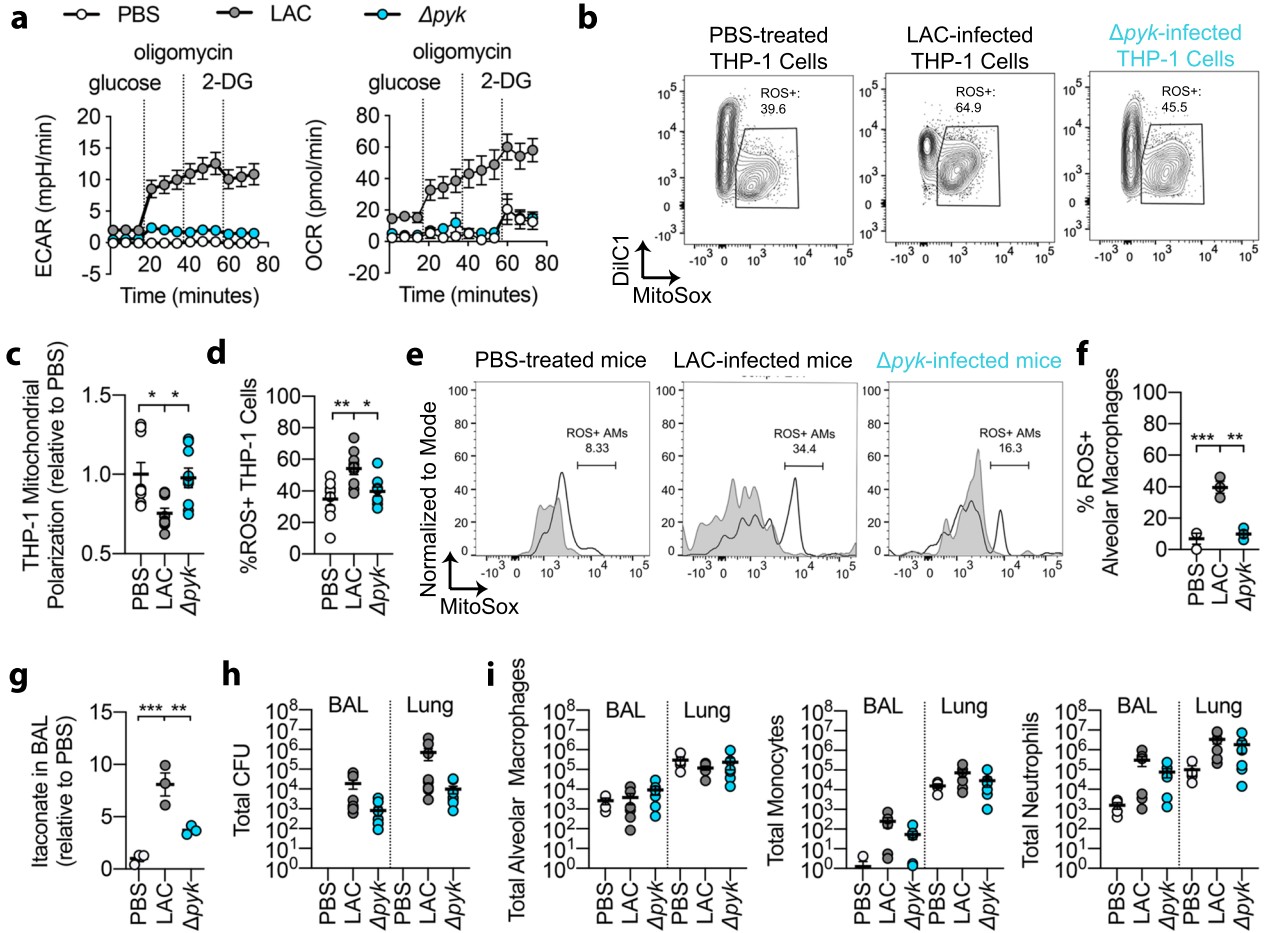

**Fig. 2 S. aureus glycolytic activity induces host mitochondrial ROS and itaconate production. a** Extracellular acidification rate (ECAR; left panel) and oxygen consumption rate (OCR; right panel) of LAC S. aureus or a glycolytically inactive (Δpyk) mutant**. b**, **c** Mitochondrial membrane polarization and **b**, **d** ROS generation in THP-1 cells treated with PBS or infected with LAC or Δpyk. **e**, **f** Mitochondrial ROS generation in the alveolar macrophages of mice treated with PBS or infected with LAC or Δpyk. **g** Itaconate in the BAL fluid of mice treated with PBS or infected with LAC or Δpyk. **h** Total colony forming units (CFU) and **i**. immune cells in the BAL fluid and lung tissue of mice treated with PBS or infected with LAC or Δpyk. Data are shown as mean ± SEM from $n = 3$ biological replicates from 3 independent experiments (**a**), 9 biological replicates from 3 independent experiments (**b**, **c**, **d**), 3 mice (**e**, **f**, **g**), or 6 mice (**h**, **i**). Significance determined by One-Way ANOVA with Tukey's Multiple Comparisons (**c**, **d**, **f**, **g**, **h**, **i**); *$P < 0.05$; **$P < 0.01$; ***$P < 0.001$.

**Itaconate triggers metabolic stress in S. aureus.** While itaconate stimulates anti-inflammatory and anti-oxidant responses in host cells, it is also bactericidal against S. aureus[16]. S. aureus responds to adverse environmental conditions by downregulating protein synthesis machinery, which is accomplished by reducing ribosomal assembly and expression of protein-folding chaperones[38–40]. To examine the acute effects of itaconate on S. aureus gene expression, we performed bulk RNA-sequencing and qRT-PCR on LAC grown in LB media with or without added itaconate. We observed global changes in the S. aureus transcriptome upon exposure to itaconate (Fig. 3a). Itaconate consistently decreased expression of loci involved in protein folding, including the chaperones dnaK and clpB (Fig. 3b, c). These changes were associated with variable expression of the 50 S ribosomal subunits rplB and rplW (Fig. 3b, c). Effects on protein synthesis pathways were coupled with dynamic changes in virulence factor mRNA levels. Transcription of the agr-controlled serine protease splA was increased in response to itaconate exposure[41], and sarR, a negative regulator of the agr locus, was also upregulated[42] (Fig. 3b, c). Interestingly, though itaconate did not affect mRNA levels of the agr-controlled alpha toxin (hla) (Fig. 3b, c), it did suppress alpha toxin production (Supplementary Fig. 3a).

Data obtained by qRT-PCR and RNA-seq yielded varying expression levels of the peroxide sensor perR, suggesting fluctuations in the repression of the oxidant stress response[43] (Fig. 3b, c). We also observed conserved expression of relA, a GTP pyrophosphokinase that synthesizes (p)ppGpp, the principal alarmone of the stringent response[38,39]. RelA is required for bacterial survival during environmental amino acid starvation[39] (Fig. 3b, c). As our itaconate experiments were performed in amino acid-rich media (LB broth), conserved relA expression was consistent with unaltered amino acid assimilation by S. aureus (Fig. 3b, c). The net effects of itaconate on S. aureus protein synthesis metabolism were reflected in its suppression of LAC growth in LB media (Fig. 3d). These data demonstrate that itaconate induces global stress in S. aureus, which is associated both with changes in protein production and alterations of other critical pathways that support planktonic proliferation.

**Itaconate alters S. aureus metabolism by inhibiting glycolysis.** Reduced protein synthesis and growth despite an abundance of carbon sources in the environment suggested that itaconate-treated S. aureus had impaired metabolism. Given that itaconate

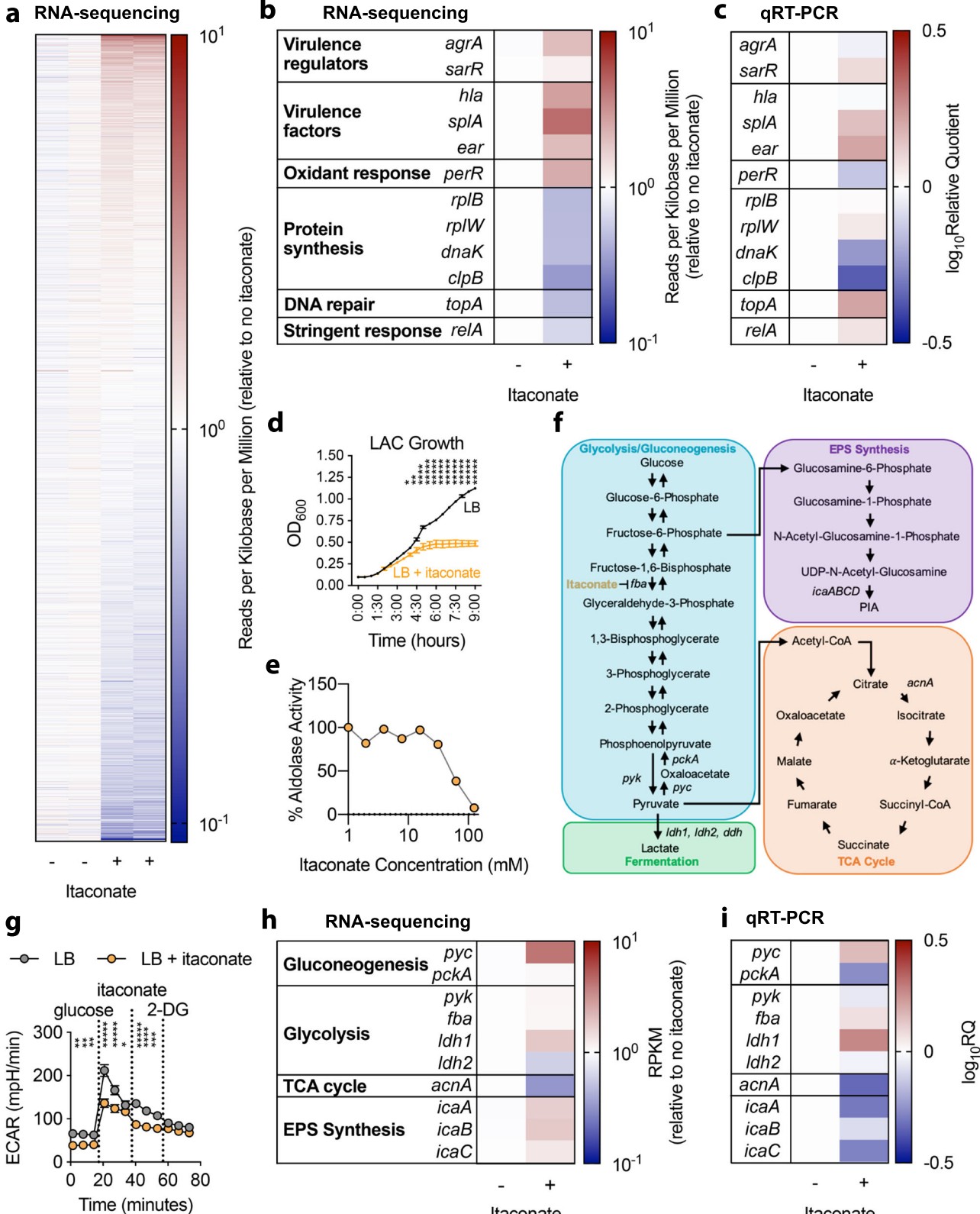

**Fig. 3 Itaconate induces *S. aureus* metabolic stress. a** Changes in the *S. aureus* LAC transcriptome upon exposure to itaconate (30 mM). **b, c** Stress response gene expression in LAC exposed to itaconate. **d** LAC growth in the presence of itaconate (30 mM). **e** Inhibition of LAC aldolase activity by itaconate. **f** Depiction of *S. aureus* central carbon metabolism. **g** Extracellular acidification rate (ECAR) of LAC grown with or without itaconate (30 mM) for 72 hours. **h, i** Central carbohydrate metabolism gene expression in LAC exposed to itaconate (30 mM). Data are shown as mean ± SEM from $n = 2$ biological replicates from one independent experiment (**a**, **b**, **e**, **h**) or 3 biological replicates from 3 independent experiments (**c**, **d**, **g**, **i**). Significance determined by two-tailed t-student with FDR correction (d, g); *$P < 0.05$, **$P < 0.01$, ***$P < 0.001$, ****$P < 0.0001$, *****$P < 0.00001$.

directly inhibits glycolytic enzymes in mammalian cells[13], we hypothesized that itaconate limited *S. aureus* growth by inhibiting staphylococcal aldolase (*fba*), the glycolytic enzyme that converts fructose-1,6-biphosphate into glyceraldehyde-3-phosphate (Fig. 3f). Using protein extracts, we confirmed that itaconate inhibited LAC aldolase activity in a dose dependent manner (Fig. 3e). This glycolytic inhibition not only limited *S. aureus* planktonic growth (Fig. 3d), but also selected for strains with reduced glycolytic activity (Fig. 3g). These data indicate that itaconate induces metabolic stress in *S. aureus* by targeting glycolysis.

We postulated that this glycolytic inhibition would have several consequences in connected metabolic pathways. We expected to see downregulation of pathways downstream of glycolysis, such as the tricarboxylic acid (TCA) cycle (Fig. 3f). Indeed, we found reduced expression of aconitase (*acnA*), the enzyme that converts citrate into isocitrate in the second step of the TCA cycle (Fig. 3h, i). Importantly, reduced *acnA* expression has been associated with resistance to antibiotic therapy during *S. aureus* infection of macrophages[44]. Due to the changes exhibited in *acnA* and glycolysis, we also expected to see upregulation of pathways that would rebalance energy production within the cell. We observed increased expression of lactate dehydrogenase 1 (*ldh1*; Fig. 3h, i), which is part of the fermentative core that enables energy generation when the TCA cycle is interrupted, and maintains redox homeostasis during oxidant stress[45]. We also detected upregulation of pyruvate carboxylase (*pyc*) (Fig. 3h, i), an enzyme involved in the generation of oxaloacetate, a precursor of gluconeogenesis. Increased *pyc* expression coupled with decreased *acnA* mRNA levels indicates that *S. aureus* rebalances metabolism by accumulating metabolites that may participate in the synthesis of carbohydrates (Fig. 3f). These transcriptomic changes are consistent with glycolytic inhibition by itaconate, and would contribute to the accumulation of glucose units that might be used by LAC for the generation of protective EPS.

**Itaconate diverts carbohydrate flux in *S. aureus* to promote EPS and biofilm production**. The findings described above suggest that itaconate selects for *S. aureus* strains with reduced glycolytic activity and increased carbohydrate production. We hypothesized that these carbohydrates are shunted into EPS production to support biofilm formation. To confirm the metabolic changes predicted by transcriptomics and determine if itaconate induces EPS production, we performed [13]C-glucose labeling and stable isotope tracing in LAC exposed to itaconate. We observed increased conservation of [13]C-labeled glucose (Fig. 4a, upper left panel), which was consistent with reduced glucose catabolism. We also detected substantial incorporation of [13]C-glucose into the EPS building block, UDP-N-acetyl-glucosamine, as evidenced by increased labeling of its 15, 16, and 17 carbon isotopologues (Fig. 4a, bottom panel). This augmented UDP-N-acetyl-glucosamine synthesis was also apparent in the carbon labeling of its precursors. UDP-N-acetyl-glucosamine is synthesized from uridine-triphosphate (UTP) and N-acetyl-glucosamine-1-phosphate, which is derived from fructose-6-phosphate, glutamine, and acetyl-coA (Fig. 3f). We observed increased incorporation of [13]C-glucose into the 6 and 8 carbon isotopologues of N-acetyl-glucosamine-1-phosphate (Fig. 4a, lower left panel), which correlated with rapid depletion of the 6-carbon isotopologue of fructose-6-phosphate (Fig. 4a, middle left panel). Similarly, we detected greater incorporation of [13]C-glucose into UTP (Fig. 4a, lower right panel) and its precursors, including uracil, uridine, uridine-5-monophosphate, uridine-5-diphosphate and uridine-diphosphate glucose (Supplementary Fig. 4a). While [13]C-incorporation was increased in other

pyrimidines like cytosine, thymine, and their precursor orotate (Supplementary Fig. 4b), it was not in increased in purines like adenosine and guanosine (Supplementary Fig. 4c), suggesting that itaconate specifically induces pyrimidine synthesis, which supports EPS production.

Itaconate also caused global restructuring of *S. aureus* central carbon metabolism to support increased UDP-N-acetyl-glucosamine synthesis. As predicted by the transcriptomic data, we detected increased carbon flux through intermediates that sustain or replenish gluconeogenesis, including pyruvate and phosphoenolpyruvate, as well as intermediates that can be converted to oxaloacetate, like malate and fumarate (Supplementary Fig. 5a). We also observed decreased flux through intermediates in pathways that compete with gluconeogenesis, including palmitate in the fatty acid biosynthesis pathway and hydroxymethylglutarate in the ketone synthesis pathway (Supplementary Fig. 5a). Importantly, we found increased [13]C-incorporation into citrate, but not downstream TCA cycle intermediates like cis-aconitate, α-ketoglutarate, and succinate (Supplementary Fig. 5a). This citrate accumulation is consistent with the reduced *acnA* expression we observed by RNA-Seq and qRT-PCR (Fig. 3h, i), and may reinforce the itaconate-induced changes in staphylococcal metabolism, given that citrate inhibits the glycolytic enzyme phosphofructokinase[46,47].

All of these metabolic changes resulted in greater biofilm production with increasing concentrations of itaconate (Fig. 4b), confirming that itaconate-induced EPS production supports biofilm formation. This process is supported by gluconeogenesis, as evidenced by reduced biofilm production by a phosphoenolpyruvate carboxykinase (*pckA*) mutant (Supplementary Fig. 6a). Mutants that are unable to produce polysaccharide intercellular adhesin (PIA) from UDP-N-acetyl-glucosamine due to deficiencies in the *icaABC* loci were still able to produce biofilm in the presence of itaconate (Supplementary Fig. 7a), suggesting that itaconate-induced biofilm formation is not limited to the production of a specific EPS and is instead the culmination of global metabolic changes.

**Clinical *S. aureus* isolates exhibit changes in carbohydrate metabolism**. To evaluate the clinical relevance of these itaconate-induced metabolic changes, we studied a longitudinal collection of *S. aureus* strains isolated over 15 years from a patient with CF (A2001-T2015). Given that these strains were chronically exposed to airway metabolites, we anticipated they would recapitulate some of the metabolic changes selected by itaconate. Using whole genome sequencing, we mapped sites of non-synonymous mutations (NSMs) using a strain from the same patient as a reference (Fig. 5a; loci that were not sites of mutations are not represented). Compared with early isolates, such as the A2001 strain, later mutants, like the T2015 strain accumulated NSMs in pro-glycolytic genes, including *pyk*, *acnA* and *ldh2* (Fig. 5a). These mutations were not associated with significant (*P* > 0.05) changes in mRNA transcript levels compared to LAC (Fig. 5b). Neither early nor late isolates exhibited NSMs in loci involved in gluconeogenesis, including the *pyc* and *pckA* loci. Furthermore, *pyc* and *pckA* mRNA levels were conserved in all CF *S. aureus* strains (Fig. 5b). NSMs were also identified in downstream pathways associated with consumption of glycolytic end-products. These included *mleS*, a locus involved in fermentation, and *atpF* and *menG*, two genes that are critical for oxidative phosphorylation (OXPHOS) and thus promote flux through the TCA cycle and glycolysis (Fig. 5a). The *ica* loci involved in EPS biosynthesis were not mutated and were significantly (*P* < 0.05) upregulated in both early and late isolates compared to LAC (Fig. 5b). Expression of the *ica* loci was more

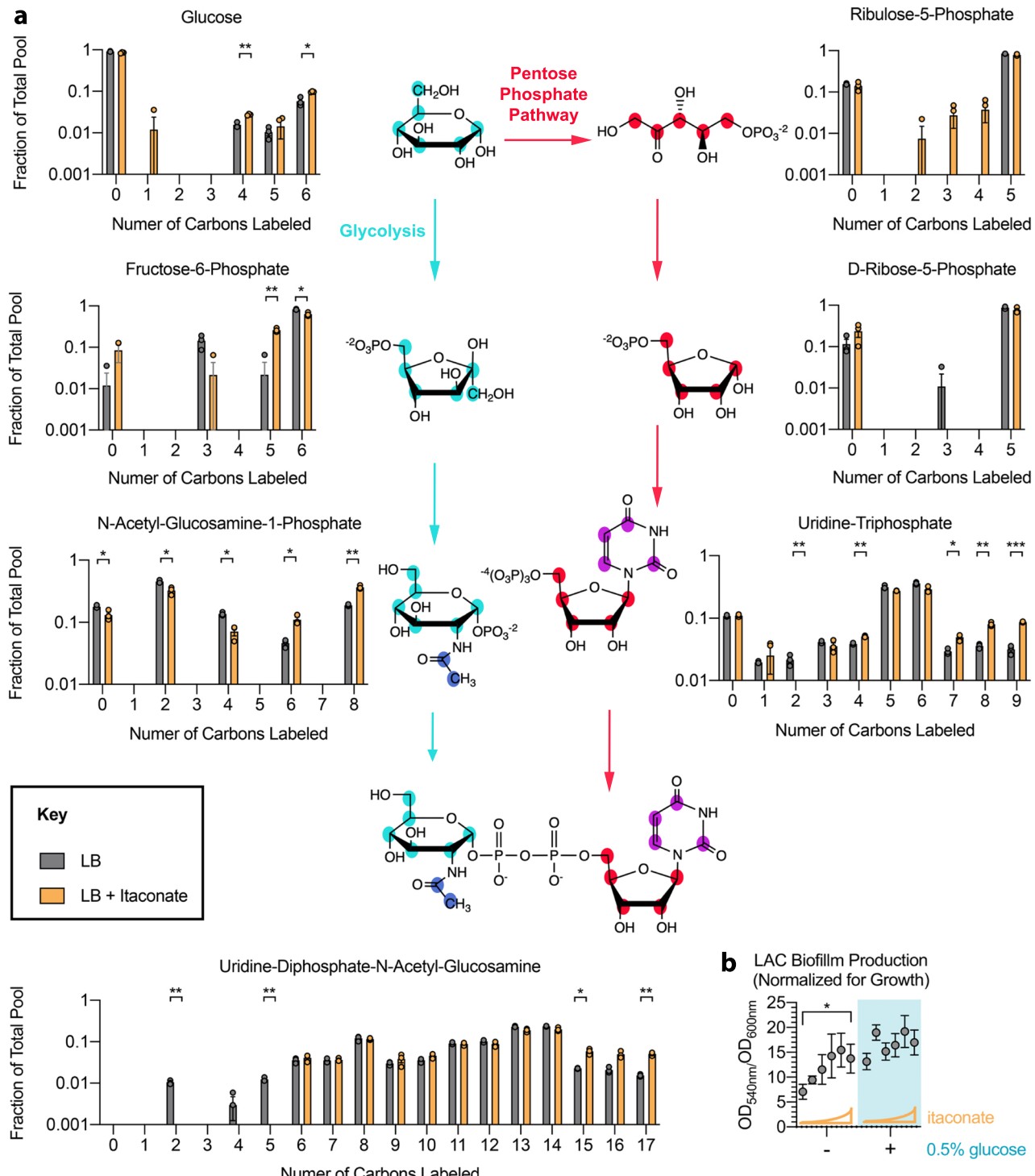

**Fig. 4 Itaconate shunts carbohydrates into production of pro-biofilm EPS in *S. aureus*. a** 13C-glucose labeling of *S. aureus* LAC metabolites involved in EPS production in the presence or absence of itaconate (30 mM). For each molecule, the different isotopologues are shown. **b** Biofilm production (normalized for growth) of LAC in increasing itaconate concentrations (0 to 62 mM), with or without glucose (0.5%). Data are shown as mean ± SEM from $n = 3$ biological replicates from one independent experiment (**a**) or 3 independent experiments (**b**). Significance determined by two-tailed t-Student with FDR correction (**a**) or One-Way ANOVA with Tukey's Multiple Comparisons (**b**); *$P < 0.05$, **$P < 0.01$, ***$P < 0.001$.

upregulated in the later mutants (Fig. 5b), representing an additional adaptive mechanism that promotes biofilm production. Overall, we observed genetic and transcriptional changes in pathways controlling central carbohydrate metabolism and extracellular polysaccharide synthesis in *S. aureus* isolates exposed to the inflamed CF lung.

**Genomic changes in the *S. aureus* isolates are consistent with adaptation to airway stress.** We observed additional adaptations that were consistent with itaconate-induced changes. The later *S. aureus* isolates exhibited more NSMs in genes associated with amino acid catabolism (e.g. *glnA*2, *proC*, *oppD*2) (Fig. 5a), protein synthesis (e.g. *alaS* and *rimP*) (Fig. 5a), RNA

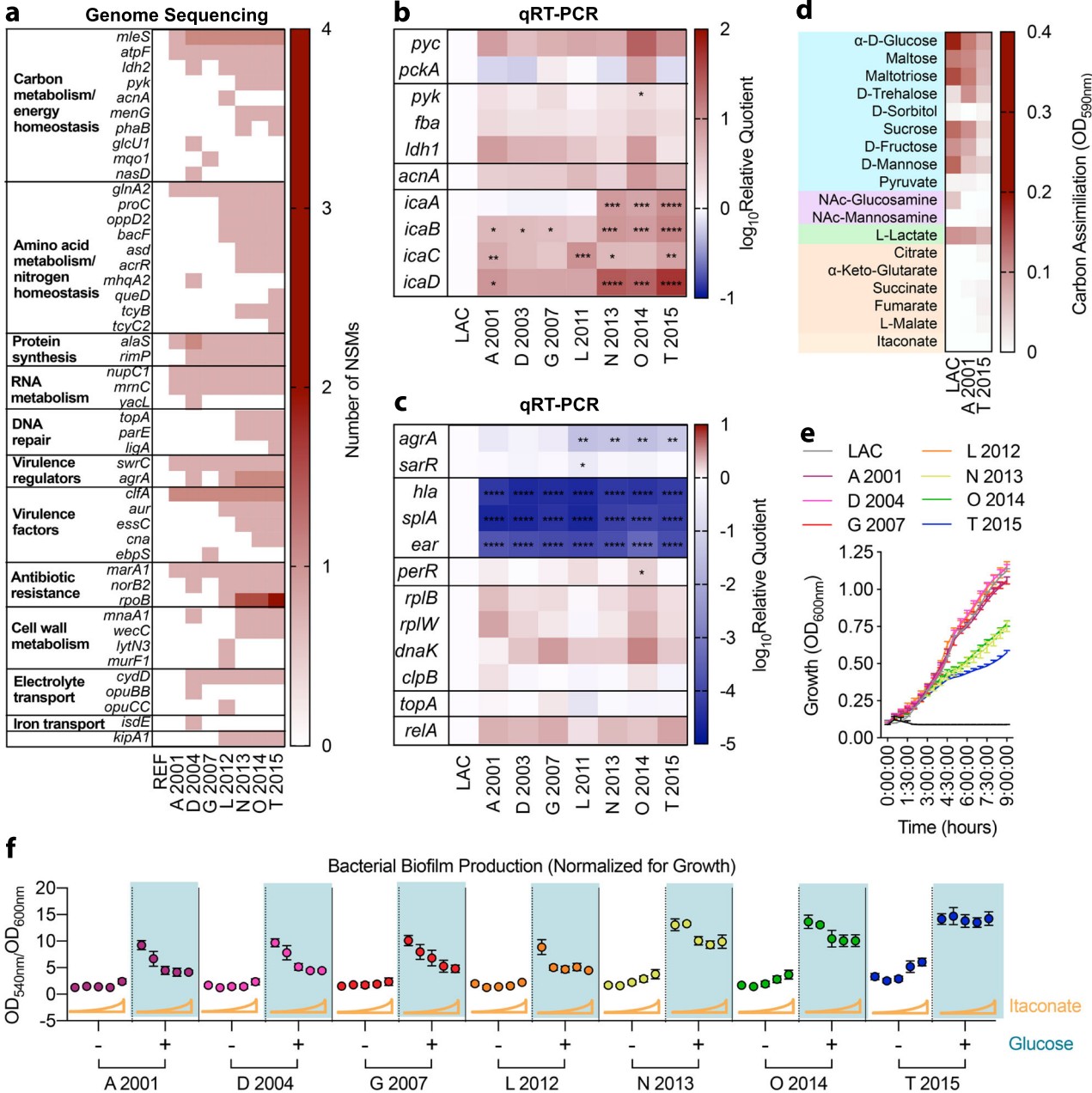

**Fig. 5 Clinical isolates exhibit adaptive metabolic changes to itaconate. a** Non-synonymous mutations (NSMs) in the genomes of clinical isolates. REF represents a reference isolate from the same patient. **b** Expression of central carbohydrate metabolic genes and **c** stress response genes by the clinical isolates, with respect to LAC. **d** Utilization of carbon sources that feed glycolysis (blue), EPS synthesis (purple), fermentation (green), or the TCA cycle (orange) by the earliest (A 2001) and latest (T 2015) clinical isolates. **e** Growth of clinical isolates in LB. **f** Biofilm production (normalized for growth) of the clinical isolates in increasing itaconate (0 to 62 mM), with or without glucose (0.5%). Data shown as mean ± SEM from $n = 3$ biological replicates from 3 independent experiments (**b**, **c**, **d**, **e**, **f**). Significance determined by One-Way ANOVA with Tukey's Multiple Comparisons (**b**, **c**); *$P < 0.05$, **$P < 0.01$, ***$P < 0.001$, ****$P < 0.0001$.

metabolism (e.g. *nupC1*, *mrnC* and *yacL*) (Fig. 5a), and DNA repair (e.g. *topA*, *parE* and *ligA*) (Fig. 5a). Of note, *relA* was not mutated and its expression was conserved in all isolates (Fig. 5c), suggesting amino acid availability and sustained production of the alarmone (p)ppGpp. Although there were no NSMs in *splA* or *hla*, these genes were significantly downregulated in the whole collection of clinical strains (Fig. 5c). The *agrA* locus, a master regulator of toxin expression induced by glucose[48] and the glycolytic end-product pyruvate[49], was mutated in most strains and significantly downregulated in the later isolates (Fig. 5a, c). Other virulence factors and their

regulators, such as *swrC*, *clfA*, *aur*, *essC* and *cna* also accrued NSMs in the later isolates (Fig. 5a), suggesting that reduced toxin activity may provide a survival advantage for *S. aureus* in the airway. Genes associated with protection against oxidant damage, such as *perR*, were conserved in these isolates, as were protein chaperones like *dnaK* and *clpB*, and the 50 S ribosomal subunits *rplB* and *rplW* (Fig. 5c). As expected, the later isolates exhibited higher numbers of NSMs in genes such as *marA1*, *norB2* and *rpoB*, which is indicative of antibiotic resistance (Fig. 5a). After chronic exposure to the oxidant-rich CF airway, *S. aureus* exhibits adaptations that are consistent with itaconate

stress, including mutations in and diminished expression of loci involved in protein synthesis and toxin production.

## *S. aureus* isolates demonstrate selection for increased biofilm production.

Given that *S. aureus* adaptation to the itaconate-rich, inflamed CF lung involved genetic and transcriptional changes in the genes associated with energy metabolism, we hypothesized that the isolates would exhibit altered assimilation of carbohydrates and TCA cycle intermediates. Moreover, we postulated that these adaptive changes would result in increased biofilm production. While the early A2001 and late T2015 clinical isolates still generated more ATP with carbohydrates than with other nutrients, such as pyruvate, citrate, α-ketoglutarate, succinate, fumarate or malate, they generated less ATP than LAC in all carbon sources tested (Fig. 5d). The diminished metabolism of the isolates was reflected in their reduced growth rates in LB media (Fig. 5e). As expected, in the presence of glucose-supplemented media, the later isolates demonstrated increasingly proficient biofilm production in the presence of itaconate (Fig. 5f). These observations indicate that *S. aureus* isolates from chronic airway infection adapt to metabolites in the CF lung environment by reducing glycolytic activity and increasing biofilm production.

## Discussion

The success of *S. aureus* as a human pathogen is due to its ability to respond to and evade the host immune response[4]. Therefore, it is not surprising that *S. aureus* strains associated with persistent pulmonary infection exhibit adaptation to airway immunometabolites. Distinguishing these Gram-positive bacteria from Gram-negative pathogens that occupy the same niche in the human airway, *S. aureus* elicits the release of itaconate, but not succinate, from activated immune cells by inducing mitochondrial oxidant stress. Itaconate in turn exerts metabolic stress on the bacteria by inhibiting staphylococcal aldolase activity and the generation of ATP through glycolysis, which is the preferred metabolic pathway for *S. aureus* during many infections[19–21]. This suppression of glycolysis alters carbohydrate metabolism and selects for a discrete set of strains that conserve gluconeogenesis, EPS synthesis, and biofilm formation, as evidenced by clinical isolates that were chronically exposed to itaconate in the airways of patients with longstanding *S. aureus* lung infection. This intricate metabolic crosstalk between *S. aureus* and the host demonstrates that, depending on the tissue and the immune response, these organisms exploit different mechanisms to assimilate available nutrients and persist.

Our data suggest that itaconate exerts its influence on *S. aureus* over time, acting as a metabolic pressure that selects for bacteria with altered genomic, transcriptomic, and metabolic profiles that support persistent infection. The acute effects of itaconate on LAC were less apparent at the transcriptional level, as evidenced by differences in mRNA expression observed by RNA-seq and qRT-PCR, as well as the discrepancy between *hla* expression and alpha toxin protein production. This is likely because itaconate instead exerts its effects at the protein level by directly inhibiting staphylococcal aldolase and altering carbon flux within the cell. Further studies are required to determine if itaconate promotes post-translational modifications in other *S. aureus* proteins, as it does in the host[13]. The impact of itaconate exposure on the staphylococcal genome and transcriptome was more evident in the host-adapted clinical isolates, which accrued NSMs in specific metabolic pathways and exhibited transcriptional changes in key metabolic and toxin genes. The significantly decreased expression of *hla* in the clinical isolates reproduces phenotypes described in other settings of chronic pneumonia[50], which, as suggested here, might be better equipped to cope with the metabolic stress exerted by itaconate.

It has long been appreciated that biofilms are associated with intractable infection in vivo. Biofilms are not just relevant to implanted device-associated infections, such as ventilator-associated pneumonia[51], but also chronic infections in patients with preexisting lung inflammation. A hallmark of chronic staphylococcal infections of the CF lung is the selection of strains that exhibit increased capacity for biofilm formation[19], suggesting that generation of these bacterial communities plays a key role in their long-term survival in the airway. The findings of this study suggest that these biofilm communities might form as a consequence of the metabolic immune response, perpetuating the persistence of *S. aureus* in the human lung rather than promoting its clearance. While ostensibly a protective response to metabolic stress evoked by itaconate, changes in *S. aureus* carbon flux and EPS synthesis are associated with a comprehensive alteration of the metabolic and transcriptomic activity of these organisms. These adaptive changes in *S. aureus* substantially differ from the ones observed in *P. aeruginosa*, which also exploits itaconate to chronically colonize the CF lung, but by metabolizing itaconate to produce energy[10]. Itaconate-adapted *S. aureus* strains cannot consume itaconate and, instead, they react to it by producing more EPS, reducing ribosomal function and protein synthesis, and diminishing virulence factor production, consistent with the "persister phenotype"[52]. This metabolic response to itaconate has profound implications for the survival of the *S. aureus* community, enabling adaptation to potentially toxic immunometabolites, and tolerance to clinically available bactericidal therapies.

IRG1 and itaconate have major immunological consequences for the host response as well. In macrophages exposed to LPS, succinate oxidation stimulates IL-1β and IL-6 proinflammatory responses, while itaconate release directly counters these pathways and activates other anti-inflammatory responses controlled by *Nrf2*[53]. In contrast, itaconate but not succinate, was the predominant immunometabolite induced by *S. aureus*, and these organisms stimulated less secretion of IL-1β and IL-6 than LPS-expressing *P. aeruginosa*. While *S. aureus* glycolytic activity was sufficient to initiate IRG1 function, further studies will be required to evaluate the mechanism by which bacterial glycolysis induces mitochondrial stress. Moreover, *S. aureus* glycolysis is apparently non-essential for *S. aureus* persistence in vivo during lung acute and chronic infections. We were surprised to find that the Δ*pyk* mutant and the host-adapted strains, which had substantially less glycolytic activity, persisted in the lung, given that in other tissues, such as the skin[35] or bone[17], *S. aureus* glycolysis is basically essential for infection. Regardless, host sensing of *S. aureus* metabolic activity, in the absence of LPS stimulation, drives a fundamentally distinct immune response in the lung.

Quantification of immunometabolite concentration on the surface of airway cells in vivo remains a major technical challenge. Untargeted metabolomics of mouse BAL fluid was performed after administrating saline, which diluted the metabolites in the airway surface liquid (ASL) layer. These diluted samples yielded itaconate concentrations in the high nanomolar range, which differ from the physiological concentrations found by others by several orders of magnitude[16,54]. Itaconate levels found in clinical samples are also influenced by the amounts of saline administered to recover artificially induced sputum in the healthy subjects, and airway dehydration and sputum density in CF patients. Due to these limitations, we performed experiments with a range of itaconate concentrations in the same millimolar scale found in other studies[16,54], validating our results with WT LAC by analyzing *S. aureus* clinical isolates naturally generated in the human lung under physiological itaconate concentrations.

Our findings highlight immunometabolic differences in the innate immune response to *S. aureus* versus Gram-negative pathogens. Both host and pathogen actively respond to itaconate accumulation in the airway, which is readily induced by both Gram-positive and Gram-negative bacteria. *S. aureus* adapts to the itaconate by restructuring its metabolism. This staphylococcal metabolic reprogramming supports persistent infection in the airway, sacrificing proliferation for the establishment of biofilm and its many advantages to overall survival.

## Methods

**Human Subjects**. All human samples (healthy individuals and CF patients) were obtained from adults (22-44 years of age) in the CF program at Yale University under the Yale IRB Protocol 0102012268 and Columbia Protocols AAAR1395. Males and females are represented in an approximate 50–50% ratio. Sputum samples from healthy adult individuals were collected after nebulization with 3% hypertonic saline for five minutes on three cycles. To reduce squamous cell contamination, subjects were asked to rinse their mouth with water and clear their throat. CF subjects expectorated sputum spontaneously for our studies. Expectorated sputum samples were collected into specimen cups and placed on ice. Of the 7 CF patients that were chronically infected with *Staphylococcus aureus* (either MSSA or MRSA), 5 patients were co-infected with other agents, such as *Pseudomonas* (3 patients), *Achromobacter* (1 patient) or human influenza (1 patient). No CF patients were infected with *Aspergillus*, a known itaconate inducer. None of the CF patients studied were under CFTR modulator therapy (e.g. Kaleydeco, Orkambi). An informed consent was signed by all subjects.

**Mouse study design**. Mice aged 7-9 weeks were purchased from Jackson Laboratories (C57BL/6J, stock number 000664). These mice were housed in humidity-controlled conditions at 18-23 degrees Celsius, with 12-hour light/dark cycles. All animal studies were approved by the Columbia IACUC Protocols AAAR9406 and AABE8600, and were carried out in strict accordance with the Guide for the Care and Use of Laboratory Animals of the NIH, the Animal Welfare Act, and US federal law. Each in vivo and ex vivo experiment was performed using an equal ratio of male: female animals. Sex was not expected to influence the final results of experiments. Animals were randomly assigned to cages. Mouse health was routinely checked by an IACUC veterinarian.

**Bacterial strains and culture**. Bacterial strains used in this study are described in Supplementary Table 1 and can be obtained upon request from the referenced lab. All strains were grown at 37 °C on plates of Luria-Bertani broth (LB, Becton Dickinson (BD) #244610) with 1% agar (w/v, Sigma #400400010). Overnight cultures and subcultures were grown in LB at 37 °C with shaking. For the Δ*pyk* mutant, the LB agar plates and LB medium were supplemented with 5% sodium pyruvate (w/v, Sigma #P5280). For the Tn::*icaA*, Tn::*icaB*, and Tn::*icaC* transposon library mutants, the LB agar plates and LB medium contained 5 μg/mL erythromycin (Sigma-Aldrich #E6376). For the experiments that used LB supplemented with itaconate (Aldrich #129204), the pH of the medium was corrected to 7.0 with 10N sodium hydroxide (Fisher Scientific #SS-2661). Bacterial concentrations were quantified by optical density at 600 nm ($OD_{600}$) and confirmed by serial dilution and plating on LB agar plates.

**Mouse infection**. Mice were infected intranasally with $2 \times 10^7$ colony forming units (CFUs) of the given *S. aureus* strain in 50 μL phosphate-buffered saline (PBS). PBS was used for an uninfected control (vehicle). 16 hours after infection (unless otherwise indicated), the mice were sacrificed for bronchoalveolar lavage (BAL) fluid and lung tissue collection. Lung tissue was homogenized through 40 μm cell strainers (Falcon #352340). Aliquots of the BAL and lung homogenates were serially diluted and plated on LB agar plates to determine CFU counts. The BAL and lung homogenates were spun down and the BAL supernatant was collected for cytokine and untargeted metabolomic analysis. After hypotonic lysis of the red blood cells, the remaining BAL and lung cells were prepared for fluorescence-activated cell sorting (FACS) analysis as described below.

**Untargeted metabolomic analysis**. Metabolites in the BAL and sputum supernatants were identified and quantified by high-resolution mass spectrometry. The metabolites were extracted in a 50% methanol (Alpha Aesar #22909):water (v/v) solution. Sample runs were performed on a Q Exactive™ HF Hybrid Quadrupole-Orbitrap™ Mass Spectrometer (Thermo-Fisher) coupled to a Vanquish™ UHPLC System (Thermo-Fisher). Chromatographic separation was achieved on a Syncronis HILIC UHPLC column (2.1 mm x 100 mm x 1.7um, Thermo-Fisher) using a binary solvent system at a flow rate of 600uL/min. Solvent A, 20 mM ammonium formate pH 3.0 in mass spectrometry grade $H_2O$; Solvent B, mass spectrometry grade acetonitrile with 0.1% formic acid (%v/v). A sample injection volume of 2uL was used. The mass spectrometer was run in negative full scan mode at a resolution of 240,000 scanning from 50-750 m/z. Metabolites were identified using the known

chromatographic retention times of standards, and metabolite signals were quantified using E-Maven v0.10.0.

**Multiarray cytokine analysis**. Cytokine concentrations in BAL supernatants were quantified with the Mouse Cytokine Proinflammatory Focused 10-plex Discovery Assay (Eve Technologies), which quantifies GM-CSF, IFNγ, IL-1β, IL-2, IL-4, IL-6, IL-10, IL-12p70, MCP-1, and TNF-α using a bead-based multiplexing technology also known as addressable laser bead immunoassay.

**Flow cytometry of mouse BAL and lung cells**. For identification of immune cell populations and quantification of IRG1: mouse BAL and lung cells were stained with LIVE/DEAD stain (Invitrogen #L23105A) and an antibody mixture of anti-CD45-AF700 (BioLegend #103127), anti-CD11b-AF594 (BioLegend #101254), anti-CD11c-Bv605 (BioLegend #117334), anti-SiglecF-APC-Cy7 (BD Biosciences #565527), anti-Epcam-FITC (BioLegend #118207), anti-F4/80-Pe-Cy7 (BioLegend #123114), anti-Ly6C-Bv421 (BioLegend #128032), and anti-Ly6G-PerCp-Cy5.5 (BioLegend #127616), each at a concentration of 1/200 in PBS, for 1 hour at 4 °C. After washing, the cells were fixed, permeabilized, and intracellularly stained with a monoclonal anti-Irg1 antibody (Abcam #ab222411) at a concentration of 1/200 in permeabilization buffer, followed by washing and staining with an anti-rabbit-AF647 (Invitrogen #A31573) at a concentration of 1/500 in permeabilization buffer, for 30 minutes at room temperature. After a final wash, the cells were stored in 2% paraformaldehyde (Electron Microscopy Sciences #15714-S) until analysis on the BD LSRII (BD Biosciences) using FACSDiva v9.

For quantification of ROS in immune cells: mouse BAL and lung cells were stained with LIVE/DEAD stain (Invitrogen #L23105A), 5 μM MitoSox (Invitrogen #M36008), and an antibody mixture of anti-CD45-AF700 (BioLegend #103127), anti-CD11b-AF594 (BioLegend #101254), anti-CD11c-Bv605 (BioLegend #117334), anti-SiglecF-APC-Cy7 (BD Biosciences #565527), anti-Epcam-FITC (BioLegend #118207), anti-F4/80-Pe-Cy7 (BioLegend #123114), anti-Ly6C-Bv421 (BioLegend #128032), and anti-Ly6G-PerCp-Cy5.5 (BioLegend #127616), each at a concentration of 1/200 in PBS, for 30 min at 37 °C. The cells were then washed with PBS and immediately analyzed on the BD LSRII.

Flow cytometry was analyzed with FlowJo v10. Mouse BAL and lung cells were identified as follows: alveolar macrophages, Epcam⁻CD45⁺CD11b⁺/⁻SiglecF⁺CD11c⁺; monocytes, CD45⁺CD11b⁺SiglecF⁻CD11c⁻Ly6G⁻Ly6C⁺/⁻; neutrophils, CD45⁺CD11b⁺SiglecF⁻CD11c⁻Ly6G⁺Ly6C⁺/⁻. An example gating strategy is provided in Supplementary Fig. 8a.

**THP-1 cell culture and infection**. THP-1 cells (ATCC, TIB-202) were cultured at 37 °C and 5% $CO_2$ in THP-1 growth medium (RPMI 1640 with L-glutamine (Corning #10-040-CV), 10% heat-inactivated fetal bovine serum (hiFBS, Gibco #F0926), and 1% penicillin/streptomycin (P/S, Corning #30-002-CI)).

Two days before infection, THP-1 cells were reseeded at $1 \times 10^6$ cells/well in a 6-well tissue culture plate (Falcon #353046) in medium supplemented with 1 μM phorbol 12-myristate 13-acetate (PMA, Sigma-Aldrich #P1585). The medium (without PMA) was refreshed the next day. 3 hours before infection, the medium in wells reserved for infection (and uninfected controls) was exchanged for medium without P/S. Cells in wells reserved for counting were trypsinized with TripleExpress (Gibco #12604-021) and counted using trypan blue stain (Invitrogen #T10282). For infection, the cells were incubated 3 hours at 37 °C at a multiplicity of infection (MOI) of 10 using bacterial suspensions in PBS. PBS alone was added to uninfected control wells.

**BMDM cell culture and infection**. BMDMs were isolated from C57BL/6 mice by surgically removing the femurs and tibias, sterilizing the bone exterior with 70% ethanol, and removing the bone marrow by flushing with PBS. The cell suspension was centrifuged for 6 minutes at 500 xg and resuspended in ACK lysis buffer to remove the red blood cells. The lysis solution was quenched with PBS, and the cells were centrifuged again and resuspended in BMDM growth medium (DMEM (Corning #10-013-CV) with 10% hiFBS and 1%P/S) supplemented with 20 ng/mL rM-CSF (PeproTech #315-02). The rM-CSF-supplemented media was replenished 3 days after isolation, and the cells were used within 7 days.

On the day of infection, the BMDMs were mechanically detached, centrifuged, and resuspended in antibiotic-free BMDM medium, and counted using trypan blue stain. For infection, the cells were incubated in suspension for 3 hours at 37 °C at a multiplicity of infection (MOI) of 10 using bacterial suspensions in PBS. PBS alone was added to uninfected control wells.

**Flow cytometry of THP-1 cells and BMDMs**. For mitochondrial depolarization and ROS detection: after the 3-hour infection, the medium was exchanged for THP1 or BMDM growth medium with 500 ng/mL gentamicin (no Pen/Strep). The cells were incubated for a 1 hour at 37 °C, then trypsinized and stained with LIVE/DEAD, 5 μM MitoSox (Invitrogen #M36008), and 50 nM DilC1 (Invitrogen #M34151) in growth media with gentamicin. After washing, the cells were analyzed immediately on BD FACSCanto II (BD Biosciences). An example gating strategy is provided in Supplementary Fig. 8b.

For IRG1 detection: after the 3-hour infection, the medium was exchanged for RPMI supplemented with hiFBS and 500 ng/mL gentamicin, with or without

500 μM MitoTempo (Sigma-Aldrich #SML0737). The cells were incubated overnight at 37 °C, then trypsinized and extracellularly stained with LIVE/DEAD. The cells were fixed and permeabilized, stained intracellularly for IRG1 (described above), and analyzed on BD LSRII. An example gating strategy is provided in Supplementary Fig. 8c.

**Extracellular flux analysis.** The XFe24 sensor cartridge (Agilent #102340-100) was calibrated as per the manufacturer's instructions overnight at 37 °C without CO₂. 500 μL of XF base medium (Agilent #102353-100) supplemented with 2 mM glutamine (LifeLine #LS-1031) was added to each well of a Seahorse XF24 well plate (Agilent #102340-100) and inoculated with $3 \times 10^7$ bacteria for a 3-hour incubation at 37 °C. The OCR and ECAR were again measured on a Seahorse XFe24 Analyzer (Agilent Technologies) using Seahorse Wave Desktop v2.6.0. Glucose (Sigma #G7021) was added at a final concentration of 10 mM, followed by the addition of pH-corrected itaconate to 30 μM, and 2-deoxyglucose (2-DG, Acros Organics #111980050) to 50 mM.

**Genomic analysis.** Genomic DNA was prepared from overnight cultures using the DNeasy blood and tissue kit (Qiagen #69504). Whole-genome sequencing (WGS) libraries were prepared for Illumina sequencing using the Nextera XT kit and sequenced on the MiSeq v3 reagent kit ($2 \times 300$ cycles). Long-read libraries were generated using the Rapid Barcoding Kit (Oxford Nanopore SQK-RBK004) and sequenced on an Oxford Nanopore MinION platform using a FLO-MIN106 flow cell. Nanopore reads were processed with Porechop v0.2.4 to remove barcode sequences and filtered using Mothur v1.22.2 to remove reads <5,000 bp length and with homopolymeric stretches >20 bp in length. Hybrid de novo assembly performed with SPAdes v3.10.1 and annotation with Prokka v1.12. Mobile genetic elements (MGEs) and prophage regions were identified by IslandViewer v4 and PHASTER.

**RNA-seq analysis.** LAC was grown in LB with or without 30 mM itaconate to late exponential phase. Bacterial pellets were incubated in a cell wall lysis mixture (described above) at 37 °C for 45 min. TRK lysis buffer (Omega Bio-tek #R6834-02) and 70% ethanol were added, and samples were transferred to E.Z.N.A RNA isolation columns (Omega Bio-tek #R6834-02). RNA was isolated following the manufacturer's instructions and treated with DNase using the DNAfree DNA removal kit (Fisher Scientific #AM1906). The RNA was precipitated with 0.1 volume 3 M sodium acetate (ThermoFisher #S209) and 3 volumes of 100% ethanol, recovered by centrifugation and washed with ice cold 70% ethanol. A ribosomal RNA-depleted cDNA library was prepared according to the manufacturer's instructions using the Universal Prokaryotic RNA-Seq Prokaryotic AnyDeplete kit (NuGEN #0363-32) and sequenced with Illumina HiSeq. Raw base calls were converted to fastq files using Bcl2fastqs. Filtered reads were aligned to the LAC_FPR3757 reference genome using STAR-Aligner v2.7.3a. The mapped reads were annotated for read groups and marked for duplicates using the Picard tools v2.22.3. The raw counts were quantified using Subreads:FeatureCounts v1.6.3 and processed for differential gene expression using DEseq2 in R v3.5.3.

**qRT-PCR analysis.** For RNA-isolation, bacteria were lysed as described above. TRK lysis buffer and 70% ethanol were added, and the samples were transferred to E.Z.N.A RNA isolated columns. RNA was isolated following manufacturer's instructions. DNA was selectively degraded using the DNA-free DNA removal kit and cDNA was generated using a High Capacity cDNA Reverse transcription kit (Applied Biosystems #43688-14) on a SimpliAmp thermocycler (Applied Biosystems). qRT-PCR was performed with either Power SYBR Green PCR Mastermix (Applied Biosystems #4367059) or PowerUp SYBR Green PCR Mastermix (Applied Biosystems #A25742) on a StepOnePlus Real-time PCR System (Applied Biosystems) using StepOne v2.3. Primers are listed in Supplementary Table 2. Data were analyzed using the $\Delta\Delta C_T$ method using 16S as a control housekeeping gene.

**Exoprotein isolation and western blotting.** Overnight *S. aureus* cultures were sub-cultured for 6.5 h with or without itaconate (30 mM) and standardized to an OD₆₀₀ of 8. Bacterial cells were pelleted by centrifugation at 4000 xg for 10 min. The supernatants were filtered through a 0.2-μm-pore-size filter (Nalgene) and precipitated with 10% (v/v) trichloroacetic acid (Sigma) at 4 °C overnight. The precipitated proteins were washed with 100% ethanol, air-dried, resuspended with 8 M urea (Sigma) and $2 \times$ LDS loading buffer (Life Technologies) and boiled. Proteins were separated by gel electrophoresis, transferred to PVDF membranes (Fisher Scientific #1B24001) and probed with an anti-hla antibody (Sigma #S7531) at a concentration of 1:5,000. A secondary antibody conjugated to horseradish peroxidase (Abcam #SC2357) was diluted 1:10,000. Images were visualized using a digital chemiluminescent detection imager (ProteinSimple). Original gels can be found in Source Data file, Supplementary Fig. 3a.

**Aldolase activity assay.** Aldolase Activity Assay buffer (Abcam #ab196994) was added to lysed bacterial cultures, which were then centrifuged at 10,000 xg and 4 °C. The supernatant was diluted 1:10 and transferred to a flat-bottomed, clear, 96-well plate (Falcon #353072) prepared with the Aldolase Activity Assay reagents

(Abcam #ab196994) supplemented with itaconate. Aldolase Activity Assay reagents were prepared per kit instructions. Serial dilutions of itaconate were made from pH-corrected 2.5 M stock solutions in assay buffer. After a 10-minute incubation at 37 °C, absorbance at 450 nm was determined on a SpectraMax M2 plate reader (Molecular Devices) using SoftMaxPro v7.0.3.

**C¹³-Glucose labeling and stable isotope tracing.** LAC was grown overnight in LB with or without 30 mM itaconate, then inoculated (1/100) into fresh LB with or without itaconate, supplemented with 0.5% C¹³-Glucose (Sigma #389374) and grown at 37 °C to late exponential phase. For metabolite extraction, each culture was diluted with 3 volumes of PBS and centrifuged at 2000 xg for 10 minutes at 1 °C. The supernatant was discarded and the pellet was washed with PBS. The pellet (30 μL in volume) was resuspended in a 3:1 methanol:water extraction solution and lysed with 10 freeze-thaw cycles by alternating emersion in liquid nitrogen and a dry-ice/ethanol bath. The debris was removed by centrifugation at 14,000 xg for 5 min at 1 °C and the supernatant was stored for analysis. Targeted LC/MS analysis was performed on a Q Exactive Orbitrap mass spectrometer (Thermo Scientific) coupled to a Vanquish UPLC system (Thermo Scientific). The Q Exactive operated in polarity-switching mode. A Sequant ZIC-HILIC column (2.1 mm i.d. × 150 mm, Merck) was used for separation of metabolites. Flow rate was set at 150 μL/min. Buffers consisted of 100% acetonitrile for mobile A, and 0.1% NH₄OH/20 mM CH₃COONH₄ in water for mobile B. Gradient ran from 85 to 30% A in 20 min followed by a wash with 30% A and re-equilibration at 85% A. Metabolites were identified on the basis of exact mass within 5 ppm and standard retention times. Relative quantitation was performed based on peak area for each isotopologue. All data analysis was done using MAVEN 2011.6.17.

**Growth and biofilm assays.** For the growth curves: A U-bottomed, clear 96-well plate (Greiner Bio-One #650161) was prepared with either LB or LB supplemented with 30 mM itaconate. Each well was inoculated with $1.5 \times 10^6$ bacteria. Absorbance at 600 nm was read every 30 minutes for 9 or 18 hours on the SpectraMax M2 plate reader, as the plate incubated at 37 °C with shaking.

For the growth and biofilm assay: A flat-bottomed, clear, 96-well plate was prepared with LB or LB with 0.5% glucose (w/v), supplemented with serially diluted, pH-corrected itaconate. Each well was inoculated with $1.5 \times 10^6$ bacteria, and the plate was left to incubate statically overnight at 37 °C. The next morning, absorbance at 600 nm was determined on an Infinite M200 plate reader (Tecan). To stain the biofilm, the supernatant was discarded, the plate was washed and dried, and the biofilm was fixed with 100% methanol, then stained with 1% crystal violet (w/v, Sigma #C6158). After discarding the staining solution and washing and drying the plate, the stained biofilm was resuspended in 33% acetic acid (v/v, Acros Organics #222140010). Absorbance at 540 nm was determined on the Infinite M200 plate reader using iControl v1.10.4.

**Carbon source utilization assay.** For the Carbon Source Phenotype Micro-array™ (Biolog), a stock solution of $2 \times 10^7$ bacteria/mL was prepared in 1X IF-Oa buffer (Biolog #72268) supplemented with 1X Redox Dye Mix A (Biolog #74221). 100 μL of this stock solution (delivering $2 \times 10^6$ bacteria) was added to each well of a PM1 Microplate™ (Biolog #12111) and the plate was incubated at 37 °C overnight. Absorbance was read at 590 nm on the infinite M200 plate reader.

**Reporting Summary.** Further information on research design is available in the Nature Research Reporting Summary linked to this article.

## Data availability
All data discussed in this study are presented in the published article and its supplementary files, which also include a reporting summary. Genomic data and transcriptomic data are available in the Sequence Read Archive, under BioProject accession PRJNA686110 (Figs. 3a and 5a). Metabolomic data are available in the MetaboLights database, under accession MTBLS2405 (Fig. 4a and Supplementary Figs. 4 and 5). Source data are provided with this paper.

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

## Acknowledgements

We thank Dr. Jonathan Koff for supplying the A1, A5, and A6 clinical isolates. We also thank Drs. Anthony Richardson, Tammy, Kielian, Juliane BubeckWardenburg, and Victor Torres for supplying the Δ*pyk*, Δ*ddh/ldh1/ldh2*, Δ*hla*, and Δ*lukAB/ED/SF/* Δ*hlgABC* mutants, respectively. Finally, we thank Drs. Xiaoyuan Yang and Guoan Zhang from the Proteomics and Metabolomics Core Facility at Weill Cornell Medicine for the targeted metabolomics analysis. K.L.T. is supported by the Columbia MSTP training grant 5T32GM007367. S.A.R. is supported by a CFF Postdoctoral fellowship RIQUEL9 17F0/PG008837 and a Vertex Research Innovation Award (RIA) PG010094. A.P. is supported by the NIH 1R35HL135800, Integrating Special Populations (ISP) Resource, CTSA, Columbia University GG011557-26, and CFF PRINCE18G0. B.C.K. is supported by the Interdisciplinary Center of Clinical Research (IZKF) of the Medical Faculty

Münster Kah2/016/16 and the German Research Foundation (DFG) KA2249/5-1. This publication is also supported by the NCATS-NIH, UL1TR001873, and FIS PI16/01381 from ISCIII, and the CCTI Flow Cytometry Core at CUMC is supported by the NIH S10RR027050.

## Author contributions

K.L.T. and S.A.R. proposed the central hypothesis, conducted experiments, and wrote the manuscript. T.W.F.L. and F.D. proposed the central hypothesis and conducted experiments. M.K.A., S.J.G., R.G., M.D., N.F., S.S., and M.S. conducted experiments. S.K. and C.J.B provided the human sputum samples. R.S., I.L., and A.U. obtained the RNA-sequencing data, untargeted metabolomics data, and genome sequencing data, respectively. B.C.K. provided the clinical isolates. A.S.P. proposed the central hypothesis and wrote the manuscript.

## Competing interests

The authors declare no competing interests.
