## [Peer Review File · Nature Communications]

REVIEWER COMMENTS

Reviewer #1 (Remarks to the Author):

In their manuscript, titled "Staphylococcus aureus induces an itaconate-dominated metabolic response in the lung that drives biofilm formation," Tomlinson et al investigate how host metabolites lead to physiologic adaptations of *S. aureus* that might promote persistence in the lung. Specifically, the authors find that unlike Gram negative pathogens such as *P. aeruginosa*, Staph induces a host metabolic response characterized by release of itaconate, rather than succinate. The effects of itaconate on Staph gene expression, growth, biofilm formation, and in vivo colonization are therefore the major foci of this study. The team concludes that itaconate alters gene expression in a manner expected to inhibit glycolytic flux, increase gluconeogenesis, and ultimately lead to the production of EPS through activation of ica-dependent PIA production. In line with this hypothesis, the authors discover that itaconate promotes biofilm formation, despite the fact that it inhibits growth in LB. Moreover, several host-adapted strains exhibit increased biofilm in the presence of itaconate relative to WT USA300-lineage strain LAC. Finally, host-adapted strains are further characterized with WGS and found to induce a distinct cytokine response in comparison to WT staph, and exhibit differential colonization patterns in IRG1 KO mice in an acute pneumonia model.

The manuscript is, in general, well written, and the findings will be of significant interest to the readership of Nature Communications and the greater host-pathogen research community. Although the staphylococcal pathogenesis field is experiencing an explosion of new research linking central metabolism to virulence, relatively little is known about how host metabolites lead to bacterial adaptations that might promote infection or pathogenicity. For this reason, I find this work of extremely high significance. Strengths of the paper include the comprehensive analytical methods (RNA seq, qRT-PCR, WGS, untargeted metabolomics), the use of both animal models and human isolates / samples, and testing of multiple staphylococcal strain backgrounds. The weaknesses of this work, in comparison, are relatively modest. In particular, the link between itaconate-induced metabolic gene expression changes and EPS is not well founded, nor is it clear that this is linked to the enhancement of biofilm formation. Second, it is not clear why strains adapted to the CF lung, and presumably adapted to itaconate exposure, do not survive better in vivo if the central hypothesis of this paper is that the metabolic adaptations enhance biofilm formation. I feel that these critiques can be addressed through textual revisions and a few new experimental considerations.

Major critiques:

1) The mechanistic link between itaconate exposure, glycolysis inhibition, and shunting of metabolism to EPS production is not well founded in this study. There is no change in ica gene expression in response to itaconate, and most of the proposed metabolic changes are inferred from gene expression profiles rather than targeted metabolomics. Perhaps more problematic is the established observation that most *S. aureus* strains do not require ica or PIA for biofilm formation. This is especially true for USA300 lineage strains, where the ica locus can be deleted without any effects on biofilm formation. Conceptually, this raises the question of why the link between glycolysis inhibition and PIA was felt to underpin the biofilm phenotype. In this reviewer's opinion, the results are exciting regardless of the biofilm matrix components. However, for the current conclusion to be more sound, two things need to be demonstrated: 1) That strains exposed to itaconate produce more PIA; 2) The deletion of the ica locus abrogates itaconate-induced biofilm formation.

2) In the text and Figure 3, pyc – encoding pyruvate carboxylase – is incorrectly referred to as pro-gluconeogenic, and this is the foundation for the conclusion that itaconate is inducing more gluconeogenic flux. In Figure 3F, pyc, is depicted in the wrong reaction. This enzyme converts pyruvate to OAA, not to PEP. While it is true that OAA can then be converted to PEP by pckA to fuel gluconeogenesis, there are other fates for OAA including the TCA cycle and amino acid biosynthesis. Therefore, to appropriately conclude that itaconate-induced glycolysis inhibition triggers more gluconeogenesis, the authors need to look at expression levels, and ideally flux through, PckA. I would also remove any instances in the text referring to Pyc as a gluconeogenic

enzyme.

3) In this reviewer's opinion, the statement that glycolysis is not required for lung infection (in both results section and discussion) is not well supported. Pyk mutant strains are typically attenuated in the early stages of infection, but likely not as soon as 16h post-inoculation. Moreover, it is unclear if the bacteria are even replicating in the airways of this model. To fortify this conclusion, the authors should examine later timepoints in the infection model. Otherwise the wording should be significantly changed. I suspect that the lack of a dramatic change in CFUs between the two strains is related to the lack of replication by 16hrs post-infection, and 16h is probably too early to see dramatic differences in host killing of these two strains.

4) If the central premise of this paper is that itaconate induces bacterial adaptations that promote enhanced biofilm and persistence within the host, then why do the host-adapted strains not survive better in the in vivo model? Is this because the model is only assessing the acute phase of pneumonia? Please explain in the discussion section.

Minor comments:

1) Page 3, line 11: The data do not support the wording of "distinct cytokine response" in that the same directional changes are happening, just at different magnitudes.

2) Page 4, line 24: Some discrepancies between RNAseq and RT-PCR data are highlighted. This should be discussed in the discussion section, which currently has very little critical review of conflicting data.

3) Figure 4 and 5: Consider adding a panel for biofilm formation that depicts the A540:A600 ratio

4) Page 6, line 3: ica expression does not define a "transition to the biofilm lifestyle". See major comment above and change wording.

5) Some of the metabolic wording, e.g. Page 6 Line 30-31, might be too strong for studies based solely on expression data without targeted metabolomics or metabolic flux data.

6) Page 7, line 23: "...produces more EPS than USA300" is not an appropriate statement based solely on expression data.

7) In general, the discussion could use more critical appraisal of some of the conflicting data. For example, why do the lung adapted strains survive less than USA300 in vivo in WT mice, but better in IRG1 KO mice? What are other limitations of the study? What about conflicting RNAseq and RT-PCR data?

8) Figure 1A: Consider changing from greyscale to color. The shades of grey are difficult to distinguish.

9) Figure 3E: Please add itaconate concentration to the figure legend.

10) Figure 5C: is "ear" supposed to be "aur"?

11) Figure 5F: Consider using a wider color palette. Some of the lines are difficult to distinguish.

Reviewer #2 (Remarks to the Author):

This study examines the effect of itaconate on *S. aureus* in lung infection. The net outcome from itaconate is decreased glycolysis, high EPS production and biofilm formation. There are interesting elements here but the authors need to consider the following points.

1. The authors need to discuss their findings in the context of their previous paper on

Pseudomonas and biofilm formation. In the case of that bacteria, itaconate was used as a fuel source for EPS synthesis. From my reading of the current paper, that doesn't seem to be the case here, rather there is a remodelling of metabolism to allow for gluconeogenesis to occur, which then promotes biofilm production. We therefore need a broader discussion on the role of itaconate in biofilm formation in the case of different bacteria, particularly as this applies to Cystic Fibrosis.

2. The way the paper is laid out makes the overall thrust of the work difficult to follow. Figure 1 is clear enough, in that *S. aureus* induces itaconate, with the effect somehow requiring mitochondrial ROS production. How would that induce IRG-1? Is there a role for Type I interferon here? Then in Figure 2 evidence is presented for glycolysis and mitochondria but the reader is left wondering how this relates to itaconate. We learn in Fig 3 and 4 that itaconate indeed impairs glycolysis with a possible targeting of aldolase activity as one mechanism, but how these effects on glycolysis relate to the mitochondrial effects is a puzzle. We need a better narrative here and also a better mechanistic explanation for what is going on.

3. Figure 4 shows how itaconate blocks glycolysis and promotes biofilm formation. The authors speculate on a role for gluconeogenesis. This needs to be examined by blocking gluconeogenesis with inhibitors or by knock down. Is it known using other inhibitors (eg of aldolase) that in *S. aureus* blocking glycolysis boosts biofilm formation? Controls showing that would be useful to help the reader.

4. IN Fig 6 we see that IRG1 is needed for TNF production. This seems odd - what would the mechanism be?

Reviewer #3 (Remarks to the Author):

In this study, Tomlinson et al. show that a range of *S. aureus* strains and clinical isolates can induce itaconate from macrophages and in the airway of mice, and that the induction of this metabolite contributes to the adaptation of the pathogen to the lung. Of note, the authors show that itaconate levels are also elevated in sputum of CF patients compared to healthy controls. The *S. aureus*-induced production of ROS from macrophages is shown to increase itaconate levels and is dependent on Staphylococcal glycolysis. In contrast to the groups recent paper, a *pyk* KO strain can still cause infection in the lung, suggesting that bacterial glycolysis is not essential during pneumonia. The team show that *S. aureus* grown in the presence of itaconate has reduced growth rate in vitro, with a reduction in a number of protein synthesis and toxin gene transcript levels. Itaconate leads to an increase in glucose-dependent biofilm production. Interestingly, sequential isolates from a CF patient exhibited increased biofilm production in the presence of itaconate and glucose, suggesting the CF lung selects for mutations allowing for greater biofilm formation.

The authors nicely combine in vivo models with in vitro cell lines/human macrophages and relevant clinical isolates. Overall, the manuscript is well-organized and most of the results support the conclusions. Nevertheless, there are several issues that need to be address to fully support the conclusions of the study.

General comments:

1. In the manuscripts, the authors use "USA300" to denote data acquired with the strain LAC. USA300 is a CC8/CA-MRSA lineage composed of many different strains that exhibit altered phenotypes. As such, the term USA300 is not really appropriate and the authors should instead use the strain name "LAC".

2. In Fig 1E, the various isolates tested induced varying levels of itaconate. Can you comment on why the blood isolate (A6) might induce lower levels compared to the lung isolates?

3. Can *S. aureus* induce ROS in vivo in alveolar macrophages in vivo? These data seem to be missing to complete the model.

4. As presented, the reader never gets a sense of how much itaconate is actually produced in the BAL of mice, by human monocytes, or in the sputum of CF patients. What is the physiological concentration of itaconate? What is the level of itaconate in the lungs of mice during infection? In Fig 3E what concentration of itaconate was added to the media, is it similar to what they experience in the lung? Was the physiological concentration used in the in vitro RNAseq experiment and in the growth curve data?

5. The authors mention many times, including in the title, that itaconate drives a biofilm behavior. While this is true in vitro, data supporting this in vivo is lacking. Does *S. aureus* form better biofilms in WT mice compared to the *Irg1*^{-/-} KO? Or even culture the macrophages from these mice, and add the supernatants or BAL to see if there is an *Irg*-dependent effect on biofilm formation.

6. A detailed description of the clinical isolates is lacking. What is the ST/CC of these isolates? The reader is left to think that these are all USA300 as a USA300 strain (what strain/what genome?) was used as a reference for the comparative genomics.

7. Fig 5D. The interpretation of these data seems overblown. The authors are claiming that strains lacking toxins are more resistant to itaconate than WT. How could this be? The statement on Page 6 line 21 "Reduce toxin activity might safeguards ..." is misleading. These toxins are known to impact host cells, but the authors seem to think that they also protect from itaconate, which if true, it would be remarkable. In order to fully support this statement, the authors would need to do additional assays (show growth curves, use other agents of metabolic stress). Similarly, if the authors are correct, then the *saePQRS* and *agr* isogenic mutants of LAC should also be more sensitive to itaconate as they produce very few secreted proteins. Just showing a low-resolution picture of colonies is not sufficient to substantiate this extreme statement. This whole section could be removed as it is weak and distracting from the main message of the manuscript.

8. Fig 6C. The overall trend in these figures is a lower production of cytokines in the KO mice. However, these findings and any conclusion are severely impacted by the spread of the data. Are these studies properly powered to make statements of significance and biological relevance? Overall, the data in Fig 6A-D suggest minimal differences in the response of WT and *Lrg1* to lung infection by LAC. Data from panel F further support the lack of a role for LRG1 in cytokine response to *S. aureus* pulmonary infection. In contrast to the CFU data with LAC, the data with the clinical isolates (Fig 6E) do support the authors model/conclusions. So, the overall importance of itaconate in *S. aureus* pathogenesis needs to be tuned down a bit as mice defective in *Lrg1* exhibit similar phenotypes to some strains compared to others.

10. The authors should expand the discussion to include the role of *S. aureus* biofilms in pneumonia and CF. What is known?

Minor point:

1. The heatmap in Fig A could benefit from being in color, rather than grey scale.
2. In Fig 3D it would be nice to see the *hla* KO strain included in the 6h time point.
3. Fig 5 F – please change the colors to make them more distinct, as it is hard to tell the difference between the various shades of blue.
4. Need to indicate what post hoc test was used in the multiple comparison one-way ANOVA.

Reviewer #4 (Remarks to the Author):

In this original research article by Tomlinson et al, the authors claim that *S. aureus* provokes IRG1-mediated itaconate biosynthesis in macrophages/monocytes via the microbe's glycolytic activity causing mitochondrial production of reactive oxygen species, and that itaconate in turn promotes biofilm formation in *S. aureus* at the expense of glycolysis. These findings have potential relevance to the pathology of lung diseases in which *S. aureus* is an important pathogen including cystic fibrosis. Itaconate is increasingly recognized as an important metabolite that can modulate

both host and microbe cell behavior, as the authors previously reported for *P. aeruginosa*. *S. aureus* has also previously been observed to promote IL-10 production in macrophages via lactate (doi.org/10.1038/s41564-020-0756-3), and the current research adds another dimension to its relationship with the host immune system. Therefore the research is novel and likely to be received with interest by researchers working on immunometabolism and host-microbe interactions of human lung diseases.

The research presented makes clear that *S. aureus* lung infection elicits a different metabolic response compared to *P. aeruginosa*, another important pathogen in lung diseases including cystic fibrosis. Furthermore, clinical *S. aureus* strains elicit this response and itaconate is increased in CF sputum infected with *S. aureus* compared to uninfected healthy subjects.

One claim is that *S. aureus* glycolysis causes monocyte/macrophage mitochondrial production of reactive oxygen species (ROS) which was the basis of IRG-1 upregulation and itaconate biosynthesis. It was not clear to me what aspect of *S. aureus* glycolysis was causing mitochondrial stress/ROS in monocytes and macrophages. Was this an effect of secreted lactate from glycolysis (doi.org/10.1038/s41564-020-0756-3)? *S. aureus* seemed to actually decrease BAL succinate, glucose and fumarate in Fig. 1A. Was this simply because the bacteria were easier to clear by the THP-1 cells when *pyk* was knocked out? Although not significant, there were fewer bacteria in the in vivo model when *pyk* is knocked out, as discussed in the manuscript. Also, the ROS-experiments are done in THP-1 differentiated by PMA (doi.org/10.3389/fphar.2018.00071), but primary cells were not used to test this effect as validation. (I do not think it is appropriate to refer to the THP-1 model as 'human monocytes' on page 7, lines 36-37.) Additionally, the way of showing mitochondrial ROS was 5 μ M MitoSOX which causes mitochondrial toxicity (which could artificially enhance ROS production in the context of other stressors) and shows non-mitochondrial localization at this dose in neurons (doi.org/10.1016/j.freeradbiomed.2015.05.032). Because monocytes/macrophages are capable of significant antimicrobial NADPH oxidase activity and the significant non-mitochondrial localization of MitoSOX at this concentration the role of mitochondrial ROS requires more evidence to be credible. The MitoTEMPO experiment provides some of that evidence, but I would like to see more. What was the rationale behind the 500 μ M dose for MitoTEMPO?

I would request authors to define which ROS (one or more) they believe are being measured by MitoSOX somewhere in the manuscript. ROS is an umbrella term and although I believe the authors are thinking of superoxide, they never state this. MitoSOX reacts with a number of ROS including superoxide to produce similar fluorescent products (doi:10.1016/j.freeradbiomed.2011.09.030).

The clinical sample analysis in Fig. 1F raised several questions. (Note, I did not see a human subjects section in the methods but the IRB was addressed in the Reporting Summary.) How was the sputum generated for healthy subjects and was the technique comparable to that of the CF subjects? Were the CF patients tested here chronically or acutely infected with *S. aureus*? Were they infected by other organisms such as *P. aeruginosa* (which the authors have shown also causes itaconate secretion) and *Aspergillus fumigatus* (multiple *Aspergillus* species produce itaconate)? What about CF patients who have had good control of *S. aureus* infection through the use of antibiotics, including many younger patients? What about non-CF disease controls with *S. aureus* lung infection?

The manuscript would be improved by presenting the concentrations for itaconate, succinate, fumarate and glucose instead of showing only their fold changes. Concentrations can help put various experiments better in context with one another and with model systems. For instance, do the airway samples (which in the case of BAL and certain sputum collections are diluted several fold, a limitation that needs to be kept in mind) contain concentrations of itaconate that are reflective of the model systems? I.e., aldolase inhibition was demonstrated using a high dose of itaconate in isolated protein. As itaconate is not very cell permeable without esterification (doi:10.1038/nature25986), does enough itaconate get into *S. aureus* to inhibit aldolase in the intact organism, particularly at the amounts generated in the human or mouse lung? Another instance wherein metabolite concentrations would help put other experiments in context is the 0.5% glucose used in Figs. 4-5 is comparable to a diabetic level in blood and a level that would be

very high for airway fluid normally.

The untargeted metabolomics method description is very short, without enough detail for someone to repeat the experiment. The cytokine array description is also very short and it was not clear to me what other cytokines were measured and what the results were for these. The western blotting method is also not sufficiently detailed.

I noted that G3PDH is defined as glycerol-3-phosphate dehydrogenase but it should be defined as glyceraldehyde 3-phosphate dehydrogenase.

I would suggest re-rendering Fig. 1A in color for easier viewing.

Reviewer comments

Reviewer #1: *In their manuscript, titled “Staphylococcus aureus induces an itaconate-dominated metabolic response in the lung that drives biofilm formation,” Tomlinson et al investigate how host metabolites lead to physiologic adaptations of S. aureus that might promote persistence in the lung. Specifically, the authors find that unlike Gram negative pathogens such as P. aeruginosa, Staph induces a host metabolic response characterized by release of itaconate, rather than succinate. The effects of itaconate on Staph gene expression, growth, biofilm formation, and in vivo colonization are therefore the major foci of this study. The team concludes that itaconate alters gene expression in a manner expected to inhibit glycolytic flux, increase gluconeogenesis, and ultimately lead to the production of EPS through activation of ica-dependent PIA production. In line with this hypothesis, the authors discover that itaconate promotes biofilm formation, despite the fact that it inhibits growth in LB. Moreover, several host-adapted strains exhibit increased biofilm in the presence of itaconate relative to WT USA300-lineage strain LAC. Finally, host-adapted strains are further characterized with WGS and found to induce a distinct cytokine response in comparison to WT staph, and exhibit differential colonization patterns in IRG1 KO mice in an acute pneumonia model.*

The manuscript is, in general, well written, and the findings will be of significant interest to the readership of Nature Communications and the greater host-pathogen research community. Although the staphylococcal pathogenesis field is experiencing an explosion of new research linking central metabolism to virulence, relatively little is known about how host metabolites lead to bacterial adaptations that might promote infection or pathogenicity. For this reason, I find this work of extremely high significance. Strengths of the paper include the comprehensive analytical methods (RNA seq, qRT-PCR, WGS, untargeted metabolomics), the use of both animal models and human isolates / samples, and testing of multiple staphylococcal strain backgrounds. The weaknesses of this work, in comparison, are relatively modest. In particular, the link between itaconate-induced metabolic gene expression changes and EPS is not well founded, nor is it clear that this is linked to the enhancement of biofilm formation. Second, it is not clear why strains adapted to the CF lung, and presumably adapted to itaconate exposure, do not survive better in vivo if the central hypothesis of this paper is that the metabolic adaptations enhance biofilm formation. I feel that these critiques can be addressed through textual revisions and a few new experimental considerations.

Answer: We appreciate the positive feedback from the Reviewer. We believe that the opportunity to respond to these comments have substantially improved the quality of our manuscript. Detailed answers to major and minor critiques can be found below and highlighted in the text in red.

Major critiques:

1. *The mechanistic link between itaconate exposure, glycolysis inhibition, and shunting of metabolism to EPS production is not well founded in this study. There is no change in ica gene expression in response to itaconate, and most of the proposed metabolic changes are inferred from gene expression profiles rather than targeted metabolomics. Perhaps more problematic is the established observation that most S. aureus strains do not require ica or PIA for biofilm formation. This is especially true for USA300 lineage strains, where the ica locus can be deleted without any effects on biofilm formation. Conceptually, this raises the question of why the link between glycolysis inhibition and PIA was felt to underpin the biofilm phenotype. In this reviewer’s opinion, the results are exciting regardless of the biofilm matrix components. However, for the current conclusion to be more sound, two things need to be demonstrated: 1) That strains exposed to itaconate produce more PIA; 2) The deletion of the ica locus abrogates itaconate-induced biofilm formation.*

Answer: We agree with the Reviewer’s observations and have followed their suggestions. 1) To support our findings that itaconate induces EPS production in S. aureus, we performed targeted metabolomics assays using carbon labeling and stable isotope tracing with ¹³C-glucose. We found that itaconate indeed promotes greater carbon flux through pathways that produce EPS (Page 4, Line 49 - Page 5, Line 17; Figure 4, Page 21; Supplemental Figure 4, Page 29). 2) We quantified biofilm production by ica mutants in the presence of itaconate and found that loss of the ica loci did not prevent itaconate-induced biofilm production in S. aureus (Page 5, Lines 33-36; Supplemental Figure 7, Page 32).

2. *In the text and Figure 3, pyc – encoding pyruvate carboxylase – is incorrectly referred to as pro-gluconeogenic, and this is the foundation for the conclusion that itaconate is inducing more gluconeogenic flux. In Figure 3F, pyc, is depicted in the wrong reaction. This enzyme converts pyruvate to OAA, not to PEP. While it is true that OAA can then be converted to PEP by pckA to fuel gluconeogenesis, there are other fates for OAA including the TCA cycle and amino acid biosynthesis. Therefore, to appropriately conclude that itaconate-induced glycolysis inhibition triggers more gluconeogenesis, the authors need to look at expression levels, and ideally flux through, PckA. I would also remove any instances in the text referring to Pyc as a gluconeogenic enzyme.*

Answer: We fixed our misrepresentation of pyruvate carboxylase, and now refer to it as “pyruvate carboxylase (*pyc*), an enzyme involved in the generation of oxaloacetate, a precursor of gluconeogenesis” (Page 4, Lines 43-44), and have modified our depiction of *S. aureus* metabolic pathways to show that *pyc* produces oxaloacetate, which is converted to phosphoenolpyruvate by *pckA* (Figure 3F, Page 20). Our targeted metabolomics data demonstrate increased flux through gluconeogenic precursors (including phosphoenolpyruvate, the product of *pckA*) (Page 5, Lines 18-28; Supplemental Figure 5, Page 30), and follow-up biofilm assays with a *pckA* mutant demonstrated that *pckA* is necessary for increased biofilm production in the presence of the gluconeogenic precursor pyruvate (Page 5, Lines 30-33; Supplemental Figure 6, Page 31).

3. *In this reviewer’s opinion, the statement that glycolysis is not required for lung infection (in both results section and discussion) is not well supported. Pyk mutant strains are typically attenuated in the early stages of infection, but likely not as soon as 16h post-inoculation. Moreover, it is unclear if the bacteria are even replicating in the airways of this model. To fortify this conclusion, the authors should examine later timepoints in the infection model. Otherwise the wording should be significantly changed. I suspect that the lack of a dramatic change in CFUs between the two strains is related to the lack of replication by 16hrs post-infection, and 16h is probably too early to see dramatic differences in host killing of these two strains.*

Answer: We examined a later time point (40 hours post infection) in our model and found that both the WT and the *pyk* mutant strain were undetectable in the BAL fluid but still present in the lung at similar levels (Page 3, Lines 41-43; Supplemental Figure 2C, Page 27).

4. *If the central premise of this paper is that itaconate induces bacterial adaptations that promote enhanced biofilm and persistence within the host, then why do the host-adapted strains not survive better in the in vivo model? Is this because the model is only assessing the acute phase of pneumonia? Please explain in the discussion section.*

Answer: Our acute model assesses establishment of airway infection. The *S. aureus* clinical isolates lack several toxins (e.g. *hla*) that play a role in establishing initial airway infection and demonstrate metabolic sensitivity to itaconate, including less growth and increased biofilm production. This is demonstrated in our study by showing that in the presence of itaconate (WT vs *Irg1*^{-/-} animals) we recover less CFUs from the acutely infected airway. We have added a clearer explanation of this to the results (Page 6, Line 50 – Page 7, Line 2) and discussion sections of our manuscript (Page 7, Lines 30-37).

Minor comments:

1. *Page 3, line 11: The data do not support the wording of “distinct cytokine response” in that the same directional changes are happening, just at different magnitudes.*

Answer: We removed the word distinct in our description of the cytokine response (Page 3, Line 3).

2. *Page 4, line 24: Some discrepancies between RNAseq and RT-PCR data are highlighted. This should be discussed in the discussion section, which currently has very little critical review of conflicting data.*

Answer: We addressed the inconsistencies between the RNA-Seq and qRT-PCR data in the discussion (Page 7, Lines 40-43).

3. *Figure 4 and 5: Consider adding a panel for biofilm formation that depicts the A540:A600 ratio.*

Answer: We added panels to depict the A540:A600 ratio for both LAC (Figure 4B, Page 21) and the clinical isolates (Figure 5F, Page 22).

4. *Page 6, line 3: ica expression does not define a “transition to the biofilm lifestyle”. See major comment above and change wording.*

Answer: We removed the phrase “transition to the biofilm lifestyle” and modified our wording throughout the paper to be more specific in its references to EPS production vs biofilm formation.

5. *Some of the metabolic wording, e.g. Page 6 Line 30-31, might be too strong for studies based solely on expression data without targeted metabolomics or metabolic flux data.*

Answer: To strengthen these conclusions, we added targeted metabolomics experiments as indicated in **Points 1 and 2** (Page 4, Line 49 – Page 5, Line 36; Figure 4, Page 21; Supplemental Figure 4, Page 29; Supplemental Figure 5, Page 30).

6. *Page 7, line 23: “produces more EPS than USA300” is not an appropriate statement based solely on expression data.*

Answer: We removed the phrase “produces more EPS than USA300”.

7. *In general, the discussion could use more critical appraisal of some of the conflicting data. For example, why do the lung adapted strains survive less than USA300 in vivo in WT mice, but better in IRG1 KO mice? What are other limitations of the study? What about conflicting RNAseq and RT-PCR data?*

Answer: We added more critical appraisal of conflicting data in our discussion, explaining the inconsistencies between the RNA-Seq and qRT-PCR data (Page 7, Lines 40-43) as well as the limitations of our acute pneumonia model (Page 7, Lines 32-51).

8. *Figure 1A: Consider changing from greyscale to color. The shades of grey are difficult to distinguish.*

Answer: We re-rendered Figure 1A in color (Page 18).

9. *Figure 3E: Please add itaconate concentration to the figure legend.*

Answer: We added the itaconate concentration to the figure legend for what is now Figure 3D (Page 20, Line 4).

10. *Figure 5C: is “ear” supposed to be “aur”?*

Answer: We noted “ear” correctly, as we are referring to the *Escherichia coli* Ampicillin Resistance gene at the SAUSA300_0815 locus.

11. *Figure 5F: Consider using a wider color palette. Some of the lines are difficult to distinguish.*

Answer: We changed the color palette to more easily distinguish between the isolates (Figures 5E and 5F, Page 22).

- We appreciate the time and effort made by the Reviewer handling our manuscript.

Reviewer #2: This study examines the effect of itaconate on *S. aureus* in lung infection. The net outcome from itaconate is decreased glycolysis, high EPS production and biofilm formation. There are interesting elements here but the authors need to consider the following points.

1. The authors need to discuss their findings in the context of their previous paper on *Pseudomonas* and biofilm formation. In the case of that bacteria, itaconate was used as a fuel source for EPS synthesis. From my reading of the current paper, that doesn't seem to be the case here, rather there is a remodeling of metabolism to allow for gluconeogenesis to occur, which then promotes biofilm production. We therefore need a broader discussion on the role of itaconate in biofilm formation in the case of different bacteria, particularly as this applies to Cystic Fibrosis.

Answer: As suggested by the Reviewer, we included a broader discussion about biofilms in Cystic Fibrosis and clarified the distinction between itaconate-induced biofilm production in *S. aureus* vs. *P. aeruginosa* (Page 8, Lines 1-16).

2. The way the paper is laid out makes the overall thrust of the work difficult to follow. Figure 1 is clear enough, in that *S. aureus* induces itaconate, with the effect somehow requiring mitochondrial ROS production. How would that induce IRG-1? Is there a role for Type I interferon here? Then in Figure 2 evidence is presented for glycolysis and mitochondria but the reader is left wondering how this relates to itaconate. We learn in Fig 3 and 4 that itaconate indeed impairs glycolysis with a possible targeting of aldolase activity as one mechanism, but how these effects on glycolysis relate to the mitochondrial effects is a puzzle. We need a better narrative here and also a better mechanistic explanation for what is going on.

Answer: To facilitate comprehension of our work, we added new explanations for the proposed mechanisms. These sections can be found in red throughout the text. To evaluate the role of type I interferons in the IRG1 response, we measured these cytokines in the airway of infected mice. At the time points studied, we did not detect production of IFN α and IFN β in the BAL-fluid of LAC-infected mice (Page 3, Lines 20-23; Supplemental Figure 2A, Page 27), suggesting these cytokines are not involved in itaconate synthesis. We appreciate the Reviewer's point about the mechanism connecting bacterial glycolysis and host mitochondrial oxidative stress. We are still working on elucidating this mechanism and will present our findings in another work.

3. Figure 4 shows how itaconate blocks glycolysis and promotes biofilm formation. The authors speculate on a role for gluconeogenesis. This needs to be examined by blocking gluconeogenesis with inhibitors or by knock down. Is it known using other inhibitors (e.g. of aldolase) that in *S. aureus* blocking glycolysis boosts biofilm formation? Controls showing that would be useful to help the reader.

Answer: We performed biofilm assays in a mutant that cannot perform gluconeogenesis ($\Delta pckA$). We observed that, in the presence of the gluconeogenic precursor pyruvate, *S. aureus* required *pckA* to produce biofilm (Page 5, Lines 30-33; Supplemental Figure 6, Page 31). The role of gluconeogenesis in biofilm production was further supported by our new targeted metabolomics data, which demonstrate that itaconate causes changes in carbon flux that support gluconeogenesis (Page 5, Lines 18-28; Supplemental Figure 5, Page 30).

4. IN Fig 6 we see that IRG1 is needed for TNF production. This seems odd - what would the mechanism be?

Answer: We agree with the Reviewer that the cytokine data is intriguing, given that the impact of itaconate on cytokine secretion seems to differ between Gram-Positive and LPS-expressing bacteria. However, we feel that this mechanistic question is not within the scope of this manuscript.

- We thank the Reviewer for their effort and positive comments. We feel their suggestions substantially improved the quality of our work.

Reviewer #3: In this study, Tomlinson et al. show that a range of *S. aureus* strains and clinical isolates can induce itaconate from macrophages and in the airway of mice, and that the induction of this metabolite contributes to the adaptation of the pathogen to the lung. Of note, the authors show that itaconate levels are also elevated in sputum of CF patients compared to healthy controls. The *S. aureus*-induced production of ROS from macrophages is shown to increase itaconate levels and is dependent on Staphylococcal glycolysis. In contrast to the groups recent paper, a *pyk* KO strain can still cause infection in the lung, suggesting that bacterial glycolysis is not essential during pneumonia. The team show that *S. aureus* grown in the presence of itaconate has reduced growth rate *in vitro*, with a reduction in a number of protein synthesis and toxin gene transcript levels. Itaconate leads to an increase in glucose-dependent biofilm production. Interestingly, sequential isolates from a CF patient exhibited increased biofilm production in the presence of itaconate and glucose, suggesting the CF lung selects for mutations allowing for greater biofilm formation.

The authors nicely combine *in vivo* models with *in vitro* cell lines/human macrophages and relevant clinical isolates. Overall, the manuscript is well-organized and most of the results support the conclusions. Nevertheless, there are several issues that need to be address to fully support the conclusions of the study.

General comments:

1. In the manuscripts, the authors use "USA300" to denote data acquired with the strain LAC. USA300 is a CC8/CA-MRSA lineage composed of many different strains that exhibit altered phenotypes. As such, the term USA300 is not really appropriate and the authors should instead use the strain name "LAC".

Answer: We changed all references of "USA300" to "LAC".

2. In Fig 1E, the various isolates tested induced varying levels of itaconate. Can you comment on why the blood isolate (A6) might induce lower levels compared to the lung isolates?

Answer: The A6 isolate is from an environment (blood) with a different metabolic composition than the airway. Based on the premise of our study, we predict that these environmental differences would select for strains of *S. aureus* that exhibit variations in their central metabolism and, potentially, the type of immune response they trigger. In the original paper that describes the A6 isolate (doi.org/10.1165/rcmb.2018-0389OC), we showed that A6 exhibited slower growth and increased biofilm production compared to the other A-series isolates, which agrees with the conclusions of our current work. We commented on this in the text (Page 3, Lines 9-11).

3. Can *S. aureus* induce ROS *in vivo* in alveolar macrophages *in vivo*? These data seem to be missing to complete the model.

Answer: We quantified mitochondrial ROS in alveolar macrophages from mice treated with PBS or infected with either WT LAC or the Δ *pyk* mutant. We found that LAC induces ROS in alveolar macrophages *in vivo* and that this ROS induction is not as robust during infection with the *pyk* mutant (Page 3, Lines 32-34; Figure 2E-F, Page 19).

4. As presented, the reader never gets a sense of how much itaconate is actually produced in the BAL of mice, by human monocytes, or in the sputum of CF patients. What is the physiological concentration of itaconate? What is the level of itaconate in the lungs of mice during infection? In Fig 3E what concentration of itaconate was added to the media, is it similar to what they experience in the lung? Was the physiological concentration used in the *in vitro* RNAseq experiment and in the growth curve data?

Answer: Itaconate concentrations in mouse BAL fluid reach micromolar levels, even after the dilution that occurs during the procedure. These concentrations are expected to be at least one order of magnitude less than the physiological concentrations on the airway liquid surface layer (ASL). We used 30mM of itaconate for many of our *in vitro* experiments because it represented the IC₅₀ observed for aldolase. This concentration is comparable to those used in other studies with bacteria (doi.org/10.1371/journal.ppat.1005408), measured in activated RAW264.7 macrophages (doi.org/10.1073/pnas.1218599110) and intracellularly in BMDMs to inhibit succinate dehydrogenase ([doi:10.1016/j.cmet.2016.06.004](https://doi.org/10.1016/j.cmet.2016.06.004)). As *S. aureus* responds to itaconate and has C-5 carboxylate transporters, we expect that it gets into the cell. Furthermore, a recent study showed that itaconate is more cell-permeable than previously thought (doi.org/10.1038/s42255-020-0210-0).

5. The authors mention many times, including in the title, that itaconate drives a biofilm behavior. While this is true *in vitro*, data supporting this *in vivo* is lacking. Does *S. aureus* form better biofilms in WT mice compared to the *Irg1*^{-/-} KO? Or even culture the macrophages from these mice, and add the supernatants or BAL to see if there is an *Irg*-dependent effect on biofilm formation.

Answer: We do not show biofilms *in vivo* because there are major technical limitations in preserving the biofilm structure in lung tissues. *In vivo* biofilm imaging is not commonly presented in the literature. We feel that the experiments suggested by the Reviewer using either supernatants or BAL from WT and *Irg1* deficient animals do not directly address the question of how itaconate induces biofilm production by *S. aureus*. Itaconate regulates the release of other metabolites and bactericidal factors, which might interfere with the direct effects of itaconate on *S. aureus* biofilm and thus confound the results.

6. *A detailed description of the clinical isolates is lacking. What is the ST/CC of these isolates? The reader is left to think that these are all USA300 as a USA300 strain (what strain/what genome?) was used as a reference for the comparative genomics.*

Answer: We included the ST designations for each isolate (Supplemental Table 1, Page 24). We also included a more detailed description of the reference genome used for the isolate sequencing in the methods section (Page 11, Lines 24-25), and changed Figure 5A to reflect the reference genome that was used (Page 22, Line 3).

7. *Fig 5D. The interpretation of these data seems overblown. The authors are claiming that strains lacking toxins are more resistant to itaconate than WT. How could this be? The statement on Page 6 line 21 "Reduce toxin activity might safeguards ..." is misleading. These toxins are known to impact host cells, but the authors seem to think that they also protect from itaconate, which if true, it would be remarkable. In order to fully support this statement, the authors would need to do additional assays (show growth curves, use other agents of metabolic stress). Similarly, if the authors are correct, then the *saePQRS* and *agr* isogenic mutants of LAC should also be more sensitive to itaconate as they produce very few secreted proteins. Just showing a low-resolution picture of colonies is not sufficient to substantiate this extreme statement. This whole section could be removed as it is weak and distracting from the main message of the manuscript.*

Answer: To validate this statement, we performed growth curves of *hla* and *lukAB/ED/SF/hlgABC* mutants and isogenic controls in the presence or absence of itaconate. We observed that itaconate has less of an effect on proliferation of the toxin mutants compared to WT LAC. We moved these findings to Supplemental Figure 8 (Page 33) and included them in the discussion section (Page 7, Lines 47-50).

8. *Fig 6C. The overall trend in these figures is a lower production of cytokines in in the KO mice. However, these findings and any conclusion are severely impacted by the spread of the data. Are these studies properly powered to make statements of significance and biological relevance? Overall, the data in Fig 6A-D suggest minimal differences in the response of WT and *Lrg1* to lung infection by LAC. Data from panel F further support the lack of a role for LRG1 in cytokine response to *S. aureus* pulmonary infection. In contrast to the CFU data with LAC, the data with the clinical isolates (Fig 6E) do support the authors model/conclusions. So, the overall importance of itaconate in *S. aureus* pathogenesis needs to be tuned down a bit as mice defective in *Lrg1* exhibit similar phenotypes to some strains compared to others.*

Answer: We agree with the Reviewer that the effects of host itaconate on LAC-induced inflammation might be compromised by the spread of the data. We powered our studies based on effect sizes seen in previous staph infections in the same acute pneumonia model. However, *S. aureus* clinical isolates differ from WT LAC by multiple NSMs in key toxin and metabolic loci, which are potentially linked to lack of difference in cytokine release. To put more emphasis on the chronic strains than LAC, we have modified our description of these results to first discuss the effects of *Irg1* on the clinical isolate colonization *in vivo* (Page 6, Line 43 to Page 7, Line 2; Figure 6A-C), and then LAC (Page 7, Line 3-10; Supplemental Figure 9, Page 34). We also added a more detailed explanation to the discussion (Page 7, Lines 30-37).

9. *The authors should expand the discussion to include the role of *S. aureus* biofilms in pneumonia and CF. What is known?*

Answer: We expanded our discussion of *S. aureus* biofilms in CF (Page 8, Lines 1-16).

Minor points:

1. *The heatmap in Fig A could benefit from being in color, rather than grey scale.*

Answer: We re-rendered Figure 1A in color (Page 18).

2. *In Fig 3D it would be nice to see the *hla* KO strain included in the 6h time point.*

Answer: We published this control in previous studies from our own group, using this same mutant and antibody at 6h time point (doi: 10.1038/s41564-019-0597-0). Due to manuscript re-organization and extensive amount of new data, this blot has been moved to Supplemental Figure 3 (Page 28).

3. Fig 5 F – please change the colors to make them more distinct, as it is hard to tell the difference between the various shades of blue.

Answer: We changed the color palette to more easily distinguish between the isolates (Figures 5E and 5F, Page 22).

4. Need to indicate what post hoc test was used in the multiple comparison one-way ANOVA.

Answer: All One-Way ANOVAs were run with a Tukey's Multiple Comparison test, and we have added this information to all relevant figure legends.

- We are grateful for the time and effort the Reviewer expended to help us improve our manuscript. We feel that the quality of our work has substantially improved.

Reviewer #4: *In this original research article by Tomlinson et al, the authors claim that S. aureus provokes IRG1-mediated itaconate biosynthesis in macrophages/monocytes via the microbe's glycolytic activity causing mitochondrial production of reactive oxygen species, and that itaconate in turn promotes biofilm formation in S. aureus at the expense of glycolysis. These findings have potential relevance to the pathology of lung diseases in which S. aureus is an important pathogen including cystic fibrosis. Itaconate is increasingly recognized as an important metabolite that can modulate both host and microbe cell behavior, as the authors previously reported for P. aeruginosa. S. aureus has also previously been observed to promote IL-10 production in macrophages via lactate (doi.org/10.1038/s41564-020-0756-3), and the current research adds another dimension to its relationship with the host immune system. Therefore, the research is novel and likely to be received with interest by researchers working on immunometabolism and host-microbe interactions of human lung diseases.*

The research presented makes clear that S. aureus lung infection elicits a different metabolic response compared to P. aeruginosa, another important pathogen in lung diseases including cystic fibrosis. Furthermore, clinical S. aureus strains elicit this response and itaconate is increased in CF sputum infected with S. aureus compared to uninfected healthy subjects.

1. *One claim is that S. aureus glycolysis causes monocyte/macrophage mitochondrial production of reactive oxygen species (ROS) which was the basis of IRG-1 upregulation and itaconate biosynthesis. It was not clear to me what aspect of S. aureus glycolysis was causing mitochondrial stress/ROS in monocytes and macrophages. Was this an effect of secreted lactate from glycolysis (doi.org/10.1038/s41564-020-0756-3)?*

Answer: We tested whether *S. aureus* lactate production was associated with itaconate release by using a mutant that is unable to produce lactate (Δ ddh/ldh1/ldh2). We found no differences in itaconate levels in the BAL fluid compared to the WT control strain (Page 3, Lines 36-39; Supplemental Figure 2B, Page 27).

2. *S. aureus seemed to actually decrease BAL succinate, glucose and fumarate in Fig. 1A. Was this simply because the bacteria were easier to clear by the THP-1 cells when pyk was knocked out? Although not significant, there were fewer bacteria in the in vivo model when pyk is knocked out, as discussed in the manuscript.*

Answer: Although the color map in Figure 1A indicates a slight reduction in BAL succinate, fumarate, and glucose levels in the *S. aureus* infected mice, the magnitude of the effect was minimal and insignificant ($P > 0.05$), and we do not believe it to be biologically relevant. The statistically non-significant difference in *pyk* bacterial loads seems unlikely to be responsible for the significantly decrease in itaconate levels in the murine airway (Page 3, Lines 32-43).

3. *Also, the ROS-experiments are done in THP-1 differentiated by PMA (doi.org/10.3389/fphar.2018.00071), but primary cells were not used to test this effect as validation. (I do not think it is appropriate to refer to the THP-1 model as 'human monocytes' on page 7, lines 36-37).*

Answer: We validated the LAC-induced ROS production in primary cells (BMDMs) (Page 3, Lines 17-18; Supplemental Figure 1, Page 26) and removed all references to THP-1 cells as "human monocytes".

4. *Additionally, the way of showing mitochondrial ROS was 5 μ M MitoSOX which causes mitochondrial toxicity (which could artificially enhance ROS production in the context of other stressors) and shows non-mitochondrial localization at this dose in neurons (doi.org/10.1016/j.freeradbiomed.2015.05.032). Because monocytes/macrophages are capable of significant antimicrobial NADPH oxidase activity and the significant non-mitochondrial localization of MitoSOX at this concentration the role of mitochondrial ROS requires more evidence to be credible. What was the rationale behind the 500 μ M dose for MitoTEMPO?*

Answer: Macrophages differ from neurons in their capacity to generate and tolerate oxidant stress. We chose 5 μ M MitoSox because it has been widely used to detect mitochondrial ROS in macrophages and other myeloid cells ([10.1016/j.immuni.2016.01.001](https://doi.org/10.1016/j.immuni.2016.01.001), doi.org/10.1161/CIRCRESAHA.114.302153, doi.org/10.1186/s12931-019-1196-6). We agree with the Reviewer that there is the potential for non-specific localization given the findings in other cell types, so we included the mitochondrial polarization stain (DiIc1) to further probe changes in mitochondrial physiology specifically. The combination of DiIc1 and MitoSox indicate that there is indeed specific mitochondrial dysfunction and ROS production.

We used the same MitoTEMPO dose that was used in another cell-based infection model (doi: [10.1165/rcmb.2013-0003OC](https://doi.org/10.1165/rcmb.2013-0003OC); doi.org/10.1016/j.cell.2016.08.064), which is adequate to track oxidative changes in immune cells.

5. *I would request authors to define which ROS (one or more) they believe are being measured by MitoSOX somewhere in the manuscript. ROS is an umbrella term and although I believe the authors are thinking of superoxide, they never state this. MitoSOX reacts with a number of ROS including superoxide to produce similar fluorescent products (doi:10.1016/j.freeradbiomed.2011.09.030).*

Answer: We defined mitochondrial ROS as anion superoxide ($O_2^{\cdot -}$) in the first reference to it in the Results section (Page 3, Lines 16-17).

6. *The clinical sample analysis in Fig. 1F raised several questions. (Note, I did not see a human subjects section in the methods but the IRB was addressed in the Reporting Summary.) How was the sputum generated for healthy subjects and was the technique comparable to that of the CF subjects? Were the CF patients tested here chronically or acutely infected with S. aureus? Were they infected by other organisms such as P. aeruginosa (which the authors have shown also causes itaconate secretion) and Aspergillus fumigatus (multiple Aspergillus species produce itaconate)? What about CF patients who have had good control of S. aureus infection through the use of antibiotics, including many younger patients? What about non-CF disease controls with S. aureus lung infection?*

Answer: We added a human subjects description to the Methods section (Page 9, Lines 3-13), including almost all information mentioned by the Reviewer. Unfortunately, we did not have access to sputum samples from younger patients with good control of S. aureus through antibiotic use, nor to sputum from non-CF S. aureus pneumonia patients.

7. *The manuscript would be improved by presenting the concentrations for itaconate, succinate, fumarate and glucose instead of showing only their fold changes. Concentrations can help put various experiments better in context with one another and with model systems. For instance, do the airway samples (which in the case of BAL and certain sputum collections are diluted several fold, a limitation that needs to be kept in mind) contain concentrations of itaconate that are reflective of the model systems? I.e., aldolase inhibition was demonstrated using a high dose of itaconate in isolated protein. As itaconate is not very cell permeable without esterification (doi:10.1038/nature25986), does enough itaconate get into S. aureus to inhibit aldolase in the intact organism, particularly at the amounts generated in the human or mouse lung? Another instance wherein metabolite concentrations would help put other experiments in context is the 0.5% glucose used in Figs. 4-5 is comparable to a diabetic level in blood and a level that would be very high for airway fluid normally.*

Answer: We understand the importance of this point raised by the Reviewer. Please refer to the answer to **Reviewer 3, Point 4** who also asked this important question regarding itaconate concentration. In addition, we have added a brief discussion paragraph to the current version of our manuscript (Page 8, Lines 29-35).

We selected the 0.5% glucose concentration based on previously published biofilm assay conditions (doi.org/10.1165/rcmb.2018-0389OC). However, we agree that it is a high concentration (even for CF sputum, which tends to be 3-4 mM), so we included the non-glucose condition as well.

8. *The untargeted metabolomics method description is very short, without enough detail for someone to repeat the experiment. The cytokine array description is also very short and it was not clear to me what other cytokines were measured and what the results were for these. The western blotting method is also not sufficiently detailed.*

Answer: We added more detail to the untargeted metabolomics (Page 9, Lines 40-49), cytokine array (Page 10, Lines 1-4), and western blotting methods (Page 12, Lines 2-10). For the cytokine array results, we presented all measured cytokines from the WT vs *pyk* mutant experiment in Supplemental Figure 2D (Page 27).

9. *I noted that G3PDH is defined as glycerol-3-phosphate dehydrogenase but it should be defined as glyceraldehyde 3-phosphate dehydrogenase.*

Answer: We thank the Reviewer for detecting this error. We fixed the definition of G3PDH (Page 2, Lines 18-19).

10. *I would suggest re-rendering Fig. 1A in color for easier viewing.*

Answer: We re-rendered Figure 1A in color (Page 18).

- We thank the Reviewer for their positive comments and time expended handling our manuscript. We feel the new version of our work has clarified important points, improving its overall quality.

REVIEWER COMMENTS

Reviewer #1 (Remarks to the Author):

The authors have successfully addressed all prior critiques. The manuscript is now even stronger and the data firmly support the conclusions in the revised text.

Reviewer #2 (Remarks to the Author):

The authors have adequately addressed my concerns and I am happy to recommend acceptance.

Reviewer #3 (Remarks to the Author):

The revised manuscript is much improved, but there are still a number of instances that the conclusions are overblown and the interpretation questionable. These issues have to be properly addressed.

General comments:

1. The authors mention many times that itaconate drives a biofilm behavior. While this is true in vitro, data supporting this in vivo is lacking. This was a concern by this and other reviewers and is still a concern in the revised manuscript. While it is ok to postulate that biofilm could be involved in vivo, the lack of data to directly prove this make any definitive statement superfluous.

2. There is no evidence of biofilm formation in vivo, as such the current title is misleading.

3. The manuscript contains many strong statements that need to be qualified and tuned down:

- Page 1; last sentence: "Thus, the ability of the Gram-positive bacterium *S. aureus* to adapt to the itaconate-dominated immunometabolic response in the lung contributes to chronic infection". While this is the hypothesis of this study, I am not entirely convinced that the in vivo data shown supports this conclusion.

4. In the revised manuscript, the authors explain that they have used strain LAC as a representative of a USA300 CA-MRSA strain. In page 2, 2nd sentence "The lung is a common site of *S. aureus* infection, as evidenced a decade ago by the epidemic of acute pneumonia caused by the toxin-producing LAC strain 2,3." This statement is misleading. While the lung is indeed a common site of *S. aureus* infection and USA300 strains can cause necrotizing pneumonia, the strain LAC was isolated from a skin and soft tissue infection, not a lung infection. As per their interpretation of the A6 strain, it could also be argued that LAC also has a different metabolic composition than the strains isolated from the airway.

5. The authors have decided to keep the data regarding the role of toxins on the susceptibility to itaconate (Supplemental Figure 8A-C). This is unfortunate as the presented data is not sufficiently rigorous to substantiate any claim. Moreover, the disconnect in phenotypes between the toxin mutant strains and the agr mutant strain further highlight that something is going on. Lastly, the lack of complementation studies to establish the true connection between the toxins and the observed phenotype is also a problem. Overall, these data are not of the quality needed to support any sort of conclusion.

6. The in vivo data (Fig 6 and Supplemental Fig 9) throw into question the main conclusion of the authors. The data show minimal differences in the response of WT and *Irg1*^{-/-} to lung infection by LAC. This is surprising as itaconate has a potent growth-arresting effect on LAC (Fig 3D). In contrast, the clinical isolates do better in the KO mice. Thus, a different interpretation of the in vivo data is that the clinical isolates are more susceptible to itaconate in vivo, which question the interpretation of the in vitro data and the model put forward by the authors. As mentioned in my

first review, based on these issues, the overall importance of itaconate in *S. aureus* pathogenesis needs to be tuned down.

Reviewer #4 (Remarks to the Author):

I thank the authors for responding to my first set of comments. I believe they have in many ways improved their manuscript and I remain enthusiastic about it. They have addressed many of my concerns, but I still have the following ones:

I am not yet convinced that the decrease in BAL itaconate isn't necessarily a product of decreased bacterial burden. Itaconate decreased by 2-fold when *pyk* was knocked out (Fig. 2G), and cfus in BAL and lung by greater than an order of magnitude (Fig. 2H). Whether the comparison in 2H is $p < 0.05$ or not doesn't change the large observed decrease in average colony counts. If there are more BAL that could be analyzed for itaconate from one or both of the other independent experiments reflected in 2H, could the authors then test the correlation of BAL itaconate and cfu in BAL and lung? If the correlation is weak or non-existent, it could further justify the conclusion that glycolysis, not bacterial burden, is the basis of increased itaconate production in the murine model.

I thank the authors for clarifying the details of the clinical specimens. I would revise the interpretation of Fig. 1F, which I think is too strong currently. It reads "...confirming that *S. aureus* infection is associated with accumulation of itaconate in the airway." The comparison is of CF to healthy donors with a different sputum collection method, and there are other confounding bacteria, so it's difficult to say what factor is most responsible for itaconate increase or if there are multiple factors causing it. The data do show an association with chronic infection in CF which included *S. aureus*. That Staph is sufficient to increase itaconate in the CF subjects might be demonstrated more convincingly if those 2 subjects with Staph alone were colorized differently on the dot plot (or if each subset were uniquely colored, e.g. co-i with *P.a.* given a color, *Achromobacter* another, and influenza another). This is not to suggest you are able to statistically discriminate the various sub-groups with these low numbers, but should help readers make an informed determination about the role of Staph alone within CF, vs. healthy subjects.

I repeat my request that authors present itaconate, whenever feasible, as absolute concentration instead of fold-change. I appreciate their pointing out that the result for BAL (and for induced sputum) will be more dilute than the true concentration in ASL by 10-100-fold. Having a more accurate understanding of the itaconate concentration in the ASL - extrapolating from the expected dilution factor - would be useful.

Also, because induced sputum (which is diluted compared to expectorated) is being compared to expectorated (Fig. 1F), the data need to be standardized somehow for the large expected differences in dilution, or the authors need to explain this limitation clearly in connection with these results.

POINT-BY-POINT

Tomlinson et al, 2020.

Reviewer #1:

1) *The authors have successfully addressed all prior critiques. The manuscript is now even stronger and the data firmly support the conclusions in the revised text.*

Answer: We appreciate the time and effort made by the Reviewer handling our manuscript.

Reviewer #2:

1) *The authors have adequately addressed my concerns and I am happy to recommend acceptance.*

Answer: We thank the Reviewer for investing their time in reading our manuscript.

Reviewer #3:

The revised manuscript is much improved, but there are still a number of instances that the conclusions are overblown and the interpretation questionable. These issues have to be properly addressed.

General comments:

1) The authors mention many times that itaconate drives a biofilm behavior. While this is true *in vitro*, data supporting this *in vivo* is lacking. This was a concern by this and other reviewers and is still a concern in the revised manuscript. While it is ok to postulate that biofilm could be involved *in vivo*, the lack of data to directly prove this make any definitive statement superfluous.

Answer: In the current version of our manuscript, we have made multiple changes addressing this point. We have modified our title (Page 1, lines 1-2), abstract (Page 1, lines 32-34) and discussion (Page 8, lines 22-24) to postulate that airway infection by *S. aureus* might be influenced by the pro-biofilm effects itaconate has on staphylococcal metabolism.

2) There is no evidence of biofilm formation *in vivo*, as such the current title is misleading.

Answer: We and others have faced multiple technical difficulties tracking biofilm formation *in vivo*. We expect to provide *in vivo* biofilm imaging in a future work. To address the comment made by the Reviewer, we have modified our title accordingly. These changes can be found on Page 1, lines 1-2.

3) The manuscript contains many strong statements that need to be qualified and tuned down:

• Page 1; last sentence: "Thus, the ability of the Gram-positive bacterium *S. aureus* to adapt to the itaconate-dominated immunometabolic response in the lung contributes to chronic infection". While this is the hypothesis of this study, I am not entirely convinced that the *in vivo* data shown supports this conclusion.

Answer: We have tuned down our description of the impact that itaconate has on the capacity of *S. aureus* to cause chronic disease. These changes can be found on Page 1, lines 32-34.

4) In the revised manuscript, the authors explain that they have used strain LAC as a representative of a USA300 CA-MRSA strain. In page 2, 2nd sentence "The lung is a common site of *S. aureus* infection, as evidenced a decade ago by the epidemic of acute pneumonia caused by the toxin-producing LAC strain 2,3." This statement is misleading. While the lung is indeed a common site of *S. aureus* infection and USA300 strains can cause necrotizing pneumonia, the strain LAC was isolated from a skin and soft tissue infection, not a lung infection. As per their interpretation of the A6 strain, it could also be argued that LAC also has a different metabolic composition than the strains isolated from the airway.

Answer: We have replaced "LAC" with "methicillin-resistant *S. aureus* (MRSA)". Changes can be found on Page 2, lines 4-5.

5) The authors have decided to keep the data regarding the role of toxins on the susceptibility to itaconate (Supplemental Figure 8A-C). This is unfortunate as the presented data is not sufficiently rigorous to substantiate any claim. Moreover, the disconnect in phenotypes between the toxin mutant strains and the *agr* mutant strain further highlight that something is going on. Lastly, the lack of complementation studies to establish the true connection between the toxins and the observed phenotype is also a problem. Overall, these data are not of the quality needed to support any sort of conclusion.

Answer: We have removed this data and its description from the manuscript.

6) The *in vivo* data (Fig 6 and Supplemental Fig 9) throw into question the main conclusion of the authors. The data show minimal differences in the response of WT and *Irg1*^{-/-} to lung infection by LAC. This is surprising as itaconate has a potent growth-arresting effect on LAC (Fig 3D). In contrast, the clinical isolates do better in the KO mice. Thus, a different interpretation of the *in vivo* data is that the clinical isolates are more susceptible to itaconate *in vivo*, which question the interpretation of the *in vitro* data and the model put forward by the authors. As mentioned in my first review, based on these issues, the overall importance of itaconate in *S. aureus* pathogenesis needs to be tuned down.

Answer: To avoid confusion for the readers, and as suggested by the Reviewer and Editors, we have removed Figure 6 and Supplemental Figure 9 from the manuscript. We appreciate that the CFU differences between LAC and the clinical isolates in WT and *Irg1*^{-/-} animals might seem contradictory to the conclusions about the importance of itaconate in *S. aureus* pathogenesis. We addressed in our article that *in vivo* itaconate effects on *S. aureus* metabolism are cumulative and expected to be more evident in a long-term setting of infection, and that a mouse model of acute LAC pneumonia is insufficient to prove effects on airway bacterial burden, as USA300 does not persist long enough to track the CFU differences caused by *Irg1*. Regardless, we validated our hypothesis with clinically relevant *S. aureus* isolates, which exhibited the long-term effects of this metabolic pressure in the human lung.

- We appreciate the time invested by the Reviewer in handling our manuscript.

Reviewer #4:

I thank the authors for responding to my first set of comments. I believe they have in many ways improved their manuscript and I remain enthusiastic about it. They have addressed many of my concerns, but I still have the following ones:

1) *I am not yet convinced that the decrease in BAL itaconate isn't necessarily a product of decreased bacterial burden. Itaconate decreased by 2-fold when *pyk* was knocked out (Fig. 2G), and *cfus* in BAL and lung by greater than an order of magnitude (Fig. 2H). Whether the comparison in 2H is $p < 0.05$ or not doesn't change the large observed decrease in average colony counts. If there are more BAL that could be analyzed for itaconate from one or both of the other independent experiments reflected in 2H, could the authors then test the correlation of BAL itaconate and *cfu* in BAL and lung? If the correlation is weak or non-existent, it could further justify the conclusion that glycolysis, not bacterial burden, is the basis of increased itaconate production in the murine model.*

Answer: As suggested by the Reviewer, we have plotted BAL CFUs vs BAL itaconate for WT LAC and the *pyk* mutant. We did not observe a strong correlation between both variables (Pearson $r = -0.2075$, $p = 0.4405$), suggesting that it is *S. aureus* glycolysis instead of pathogen burden that is associated with the airway *Irg1* response. These data can be now found on Page 4, line 13-15, and in Supplemental Figure 2D.

2) *I thank the authors for clarifying the details of the clinical specimens. I would revise the interpretation of Fig. 1F, which I think is too strong currently. It reads "...confirming that *S. aureus* infection is associated with accumulation of itaconate in the airway." The comparison is of CF to healthy donors with a different sputum collection method, and there are other confounding bacteria, so it's difficult to say what factor is most responsible for itaconate increase or if there are multiple factors causing it. The data do show an association with chronic infection in CF which included *S. aureus*. That Staph is sufficient to increase itaconate in the CF subjects might be demonstrated more convincingly if those 2 subjects with Staph alone were colorized differently on the dot plot (or if each subset were uniquely colored, e.g. *co-i* with *P.a.* given a color, *Achromobacter* another, and influenza another). This is not to suggest you are able to statistically discriminate the various sub-groups with these low numbers, but should help readers make an informed determination about the role of Staph alone within CF, vs. healthy subjects.*

Answer: As suggested by the Reviewer, we have indicated each infectious agent associated with each dot presented in **Figure 1F**. By examining the samples only infected with *S. aureus*, we concluded that staphylococcal infection associates with airway itaconate accumulation. However, we also conclude that other pathogens present in the sputum are confounding factors that can be associated with the *Irg1* response in presence of *S. aureus*. These explanations can now be found on Page 3, lines 23-25.

3) *I repeat my request that authors present itaconate, whenever feasible, as absolute concentration instead of fold-change. I appreciate their pointing out that the result for BAL (and for induced sputum) will be more dilute than the true concentration in ASL by 10-100-fold. Having a more accurate understanding of the itaconate concentration in the ASL - extrapolating from the expected dilution factor - would be useful.*

Answer: We have now added itaconate concentrations induced by LAC and Δpyk mutant in mice. We present data as concentrations found in BAL (recovered after 3ml PBS perfusion into the airway), as we cannot provide the physiological ASL concentration. These values can now be found on Page 3, lines 10-11 and Page 4, line 3.

4) *Also, because induced sputum (which is diluted compared to expectorated) is being compared to expectorated (Fig. 1F), the data need to be standardized somehow for the large expected differences in dilution, or the authors need to explain this limitation clearly in connection with these results.*

Answer: We have now addressed this point about different dilution factors between induced and naturally expectorated sputum in the Discussion section of our manuscript. These changes can be found on Page 9, lines 7-10.

- We thank the Reviewer for the time and effort put into reading our manuscript.